# Mechanism of B-box 2 domain-mediated higher-order assembly of the retroviral restriction factor TRIM5α

Jonathan M Wagner[1], Marcin D Roganowicz[1], Katarzyna Skorupka[1], Steven L Alam[2], Devin Christensen[2], Ginna Doss[1], Yueping Wan[1], Gabriel A Frank[3,4], Barbie K Ganser-Pornillos[1], Wesley I Sundquist[2], Owen Pornillos[1]*

[1]Department of Molecular Physiology and Biological Physics, University of Virginia, Charlottesville, United States; [2]Department of Biochemistry, University of Utah, Salt Lake City, United States; [3]The National Institute for Biotechnology in the Negev, Ben-Gurion University of the Negev, Beer-Sheeva, Israel; [4]Department of Life Sciences, Ben-Gurion University of the Negev, Beer-Sheeva, Israel

**Abstract** Restriction factors and pattern recognition receptors are important components of intrinsic cellular defenses against viral infection. Mammalian TRIM5α proteins are restriction factors and receptors that target the capsid cores of retroviruses and activate ubiquitin-dependent antiviral responses upon capsid recognition. Here, we report crystallographic and functional studies of the TRIM5α B-box 2 domain, which mediates higher-order assembly of TRIM5 proteins. The B-box can form both dimers and trimers, and the trimers can link multiple TRIM5α proteins into a hexagonal net that matches the lattice arrangement of capsid subunits and enables avid capsid binding. Two modes of conformational flexibility allow TRIM5α to accommodate the variable curvature of retroviral capsids. B-box mediated interactions also modulate TRIM5α's E3 ubiquitin ligase activity, by stereochemically restricting how the N-terminal RING domain can dimerize. Overall, these studies define important molecular details of cellular recognition of retroviruses, and how recognition links to downstream processes to disable the virus.

*For correspondence: owp3a@eservices.virginia.edu

## Introduction

TRIM5α is a restriction factor that intercepts the incoming capsids of diverse retroviruses, including HIV-1, and inhibits viral replication. The mechanism of restriction is not yet fully understood, but is primarily associated with premature termination of reverse transcription and accelerated dissociation of the viral core (*Sayah et al., 2004*; *Stremlau et al., 2004*; *2006*). TRIM5α is also proposed to be a pattern recognition receptor for retroviral capsids, in that it initiates a signaling cascade to induce type I interferon upon capsid binding (*Pertel et al., 2011*). Ubiquitin (Ub) is implicated in both the antiviral (restriction) and signaling activities of TRIM5α. In particular, TRIM5α's E3 ligase activity creates K63-linked polyUb chains. Although the functional target or targets of ubiquitination have not been established definitively, TRIM5α self-ligation correlates with the block in reverse transcription (*Campbell et al., 2015*; *Fletcher et al., 2015*; *Roa et al., 2012*), whereas unanchored chains have been proposed to mediate interferon signaling (*Pertel et al., 2011*).

Like all TRIM proteins, TRIM5α consists of an N-terminal tripartite or RBCC motif (RING, B-box 2, and coiled-coil domains), followed by a C-terminal domain (*Figure 1A*) (*Meroni and Diez-Roux, 2005*). The B-box 2 and coiled-coil domains make an integrated antiparallel dimer fold (*Goldstone et al., 2014*; *Sanchez et al., 2014*; *Weinert et al., 2015*), which acts as a scaffold that

**eLife digest** After infecting a cell, a virus reprograms the cell to produce new copies of the virus, which then spread to other cells. However, cells have evolved ways to fight back against this infection. For example, many mammalian cells contain proteins called restriction factors that prevent the virus from multiplying. The TRIM5 proteins form one common set of restriction factors that act against a class of viruses called retroviruses.

HIV-1 and related retroviruses have a protein shell known as a capsid that surrounds the genetic material of the virus. The capsid contains several hundred repeating units, each of which consists of a hexagonal ring of six capsid proteins. Although this basic pattern is maintained across different retroviruses, the overall shape of the capsids can vary considerably. For instance, HIV-1 capsids are shaped like a cone, but other retroviruses can form cylinders or spheres.

Soon after a retrovirus enters a mammalian cell, TRIM5 proteins bind to the capsid. This causes the capsid to be destroyed, which prevents replication of the virus. Previous research has shown that many TRIM5 proteins must link up with each other via a region of their structure called the 'B-box 2' domain in order to efficiently recognize capsids. How this assembly process occurs, and why it enables the TRIM5 proteins to recognize different capsids was not fully understood. Now, Wagner et al. (and independently Li, Chandrasekaran et al.) have investigated these questions.

Wagner et al. engineered short versions of a type of TRIM5 protein called TRIM5α and used a technique called X-ray crystallography to determine the structure of its B-box domain. This revealed that the B-box present in one molecule of TRIM5α can associate with the B-boxes on two other TRIM5α molecules. By working in groups of three (or trimers), the B-box domains connect several TRIM5α proteins to form a hexagonal net. The TRIM5α net matches the arrangement of the capsid proteins in the shell of the virus, which enables TRIM5α to bind strongly to HIV-1 capsids.

Wagner et al. also found that B-box trimers are flexible, which allows the TRIM5α net to adapt to the shape of the HIV-1 capsid and wrap around regions where it curves. In addition, computer modelling suggested that the B-box trimer may also enable TRIM5α to carry out the next steps in the process of disabling the virus. Further work is now needed to understand in more detail how the trimers have this effect.

organizes the upstream and downstream domains (*Figure 1B*). In TRIM5α, the C-terminal domain is a β-sandwich fold called SPRY (or PRYSPRY/B30.2), which mediates direct binding to retroviral capsids (*Biris et al., 2012*; *2013*; *Diaz-Griffero et al., 2006b*; *Kovalskyy and Ivanov, 2014*; *Sawyer et al., 2005*; *Sayah et al., 2004*; *Sebastian and Luban, 2005*; *Stremlau et al., 2006*; *Yang et al., 2012*). The L2 linker that connects the SPRY domain to the coiled-coil packs against the coiled-coil scaffold, and so in the TRIM5α dimer, two SPRY domains are oriented to bind the capsid simultaneously (*Figure 1B*) (*Goldstone et al., 2014*; *Li et al., 2014*; *Sanchez et al., 2014*; *Weinert et al., 2015*). Capsid recognition by TRIM5α is an avidity-driven interaction; that is, productive binding only occurs in context of assembled capsid and assembled TRIM5α (*Sebastian and Luban, 2005*; *Stremlau et al., 2006*). Higher-order assembly of TRIM5α requires the B-box 2 domain, which is thought to mediate three-fold symmetric interactions that connect coiled-coil mediated TRIM5α dimers into a hexagonal net (*Diaz-Griffero et al., 2009*; *Ganser-Pornillos et al., 2011*; *Javanbakht et al., 2005*; *Li and Sodroski, 2008*; *Li et al., 2016*). This hexagonal scaffold is proposed to position the SPRY domains to match the orientations – both translational and rotational – of their corresponding binding epitopes on retroviral capsids, and thereby generate powerful avidity effects that amplify very weak (millimolar level (*Biris et al., 2013*)) intrinsic affinities between the SPRY and capsid subunits.

Higher-order TRIM5α assembly is also reported to promote the E3 ligase activity of the upstream RING domain (*Pertel et al., 2011*; *Yudina et al., 2015*), which is connected to the B-box 2 domain by the L1 linker (*Figure 1A and B*). A segment of the L1 linker forms a 4-helix bundle that mediates dimerization of the RING domain, which is required for productive interactions with Ub-conjugated E2 enzymes and formation of the catalytically active Ub ligation complex (*Yudina et al., 2015*). Although the B-box 2 domain does not appear to have a direct role in catalysis, B-box/B-box

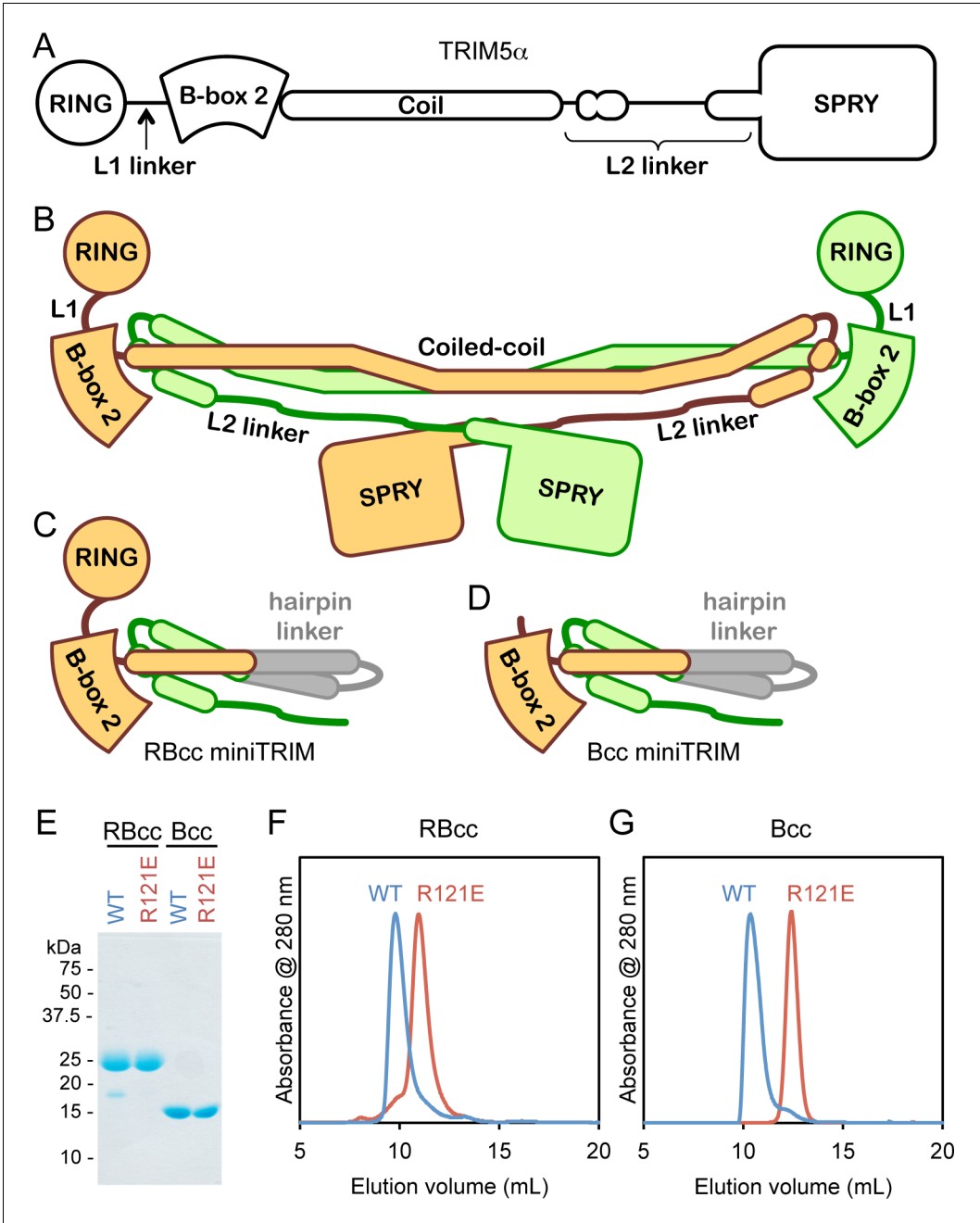

**Figure 1.** Design and oligomeric behavior of miniTRIM proteins. (**A**) Schematic of the TRIM5α primary sequence. (**B**) Schematic of the antiparallel full-length dimer. (**C–D**) Schematic of the (**C**) RBcc miniTRIM and (**D**) Bcc miniTRIM. (**E**) SDS-PAGE profiles of purified miniTRIMs. (**F–G**) Size exclusion elution profiles of (**F**) RBcc and (**G**) Bcc miniTRIMs. Wildtype (WT) constructs eluted early (blue traces), whereas R121E mutants eluted late (red traces).

The following figure supplement is available for figure 1:

**Figure supplement 1.** Primary sequence of the miniTRIMs.

interactions are expected to cluster their associated RING domains and promote RING dimerization. In context of the TRIM hexagonal lattice, the apparent juxtaposition of three-way head-to-head interactions between the B-boxes and two-way interactions between the catalytic RING domains is suggested to facilitate TRIM5α self-ubiquitination (*Yudina et al., 2015*).

The structural basis by which the B-box 2 domain promotes higher-order assembly of TRIM5α dimers has been challenging to decipher, largely because the bivalent nature of the TRIM5α dimer and the flexible architectures of both the TRIM dimer and hexagonal lattice have impeded crystallographic characterization. Analysis of the isolated B-box 2 domain has been likewise problematic, because separating the B-box from the coiled-coil exposes a 'backside' hydrophobic surface that makes it prone to aggregation. We therefore engineered artificial constructs – which we call mini-TRIMs – that retain the integrated B-box/coiled-coil fold of the full-length dimer but are more amenable to biochemical and structural analyses. These novel reagents allowed us to define the molecular details of how the B-box 2 domain facilitates higher-order assembly of TRIM5α, and how pattern recognition of retroviral capsids is coupled to ubiquitin-dependent downstream processes.

## Results

### Design and initial characterization of miniTRIMS

Our miniTRIM constructs were designed to be monovalent with respect to the RING and B-box 2 domains (to uncouple interactions mediated by these domains from the coiled-coil dimer), and yet preserve the native, quaternary B-box/coiled-coil interface (to prevent exposure of the backside hydrophobic B-box surface and non-specific aggregation). The 'RBcc' miniTRIM contained residues 1–159 from rhesus TRIM5α (which includes the RING, B-box 2, and the first 26 residues of the coiled-coil) (*Figure 1C* and *Figure 1—figure supplement 1*, colored in orange), followed by an anti-parallel coiled-coil hairpin derived from a bacterial seryl-tRNA synthetase (residues 49–78 of PDB 1SRY, gray), and then residues 225–265 of the TRIM5α coiled-coil (green). To further uncouple the B-box from potential dimeric interactions of the upstream RING domain, we also designed a second construct denoted 'Bcc' that lacks the RING (residues 1–88) (*Figure 1D*). Both the RBcc and Bcc miniTRIMs proved well behaved in solution, and could be purified to homogeneity (*Figure 1E*).

Higher-order assembly of full-length TRIM5α protein dimers requires interactions mediated by a surface patch on the B-box 2 domain that includes Arg121 (rhesus TRIM5α numbering). Biochemical and cell-based assays show that R121A and R121E mutants are deficient in capsid binding and restriction activities, and this correlates with defects in higher-order assembly (*Diaz-Griffero et al., 2009*; *Ganser-Pornillos et al., 2011*; *Li and Sodroski, 2008*). The same patch also mediates self-association of the isolated B-box 2 domain in solution (*Diaz-Griffero et al., 2009*). We therefore expected the miniTRIMs to exhibit Arg121-dependent oligomeric behavior, and we tested this by using size exclusion chromatography. Consistent with expectation, both the RBcc miniTRIM (*Figure 1F*) and Bcc miniTRIM (*Figure 1G*) eluted early from a Superdex 75 size exclusion column as asymmetric peaks with sharp leading edges and pronounced tails (blue traces). In contrast, mini-TRIMs harboring the R121E mutation eluted late as more symmetrical peaks (*Figure 1F and G*, red traces), with elution volumes consistent with monomeric species. These results indicate that the protein-protein interactions required for higher-order assembly of full-length TRIM5α are also essential for miniTRIM oligomerization.

### Bcc miniTRIM recapitulates the B-box/coiled-coil head of TRIM5α

Despite the polydisperse nature of the miniTRIMs, we obtained numerous crystal hits. We obtained high quality synchrotron diffraction data from three crystal forms of Bcc miniTRIM. A P2$_1$2$_1$2$_1$ crystal contained two trimers in the asymmetric unit (3.26 Å resolution, $R/R_{\text{free}}$ = 0.26/0.30), a C2 form contained one dimer (2.1 Å, $R/R_{\text{free}}$ = 0.18/0.22), and a P1 form contained two dimers (2.3 Å, $R/R_{\text{free}}$ = 0.22/0.26) (*Table 1* and *Supplementary file 1A*). Altogether, these yielded 12 crystallographically independent views of Bcc miniTRIM. All 12 structures were very similar to each other (*Figure 2—figure supplement 1*), and a complete structure is shown in *Figure 2A*. In general, electron densities for the B-box 2 domains and proximal regions of the coiled-coil domains were well defined (*Figure 2B*), whereas densities for the hairpin linker were of poorer quality or, in some cases, missing (*Figure 2C*). (Densities in *Figure 2B and C* are illustrated with a trimer subunit.) Thus, our structures are of high quality at the functionally relevant regions, even though the artificial linker displayed significant disorder in some cases and may not be optimally designed. Our 12 Bcc miniTRIM structures superimpose very well with the crystal structure of the TRIM5α B-box 2/coiled-coil dimer (PDB 4TN3) (*Goldstone et al., 2014*) (*Figure 2D–F* and *Supplementary file 1B*). This indicates that our

**Table 1.** Crystallographic statistics.

| | Dimer | Dimer | Trimer |
|---|---|---|---|
| *Diffraction Data* | | | |
| Beamline | APS 22ID | APS 22ID | APS 22ID |
| Wavelength (Å) | 1.0 | 1.0 | 1.0 |
| Processing program | HKL2000 | HKL2000 | HKL2000 |
| Space group | C2 | P1 | P212121 |
| Cell dimensions | $a$ = 72.7 Å | $a$ = 45.8 Å | $a$ = 71.2 Å |
| | $b$ = 41.5 Å | $b$ = 52.3 Å | $b$ = 71.5 Å |
| | $c$ = 111.3 Å | $c$ = 69.7 Å | $c$ = 213.8 Å |
| | $\alpha$ = 90°, $\beta$ = 110°, $\gamma$ = 90° | $\alpha$ = 94.8°, $\beta$ = 105.5°, $\gamma$ = 103° | $\alpha$ = 90°, $\beta$ = 90°, $\gamma$ = 90° |
| Resolution range, Å | 50-1.90 (1.97-1.90) | 50-2.30 (2.38-2.30) | 50-3.25 (3.37-3.25) |
| $R_{sym}$/$R_{meas}$ /$R_{pim}$ | 0.18(0.43) /0.12(0.90) /0.06(0.50) | 0.07(0.16) /0.10(0.23) /0.07(0.16) | 0.08(1.0) /0.05(1.0) /0.08(1.0) |
| Mean I/$\sigma$<I> | 14.0 (1.2) | 9.4 (4.0) | 26.8 (1.6) |
| Completeness,% | 98.6 (90.4) | 93.9 (80.0) | 100 (100) |
| Average redundancy | 3.5 (2.7) | 1.9 (1.7) | 13.6 (9.4) |
| Wilson B-factor, Å$^2$ | 40.5 | 36.0 | 35.1 |
| *Refinement Statistics* | | | |
| Refinement program | PHENIX | PHENIX | PHENIX |
| Resolution range | 32.5-1.91 (1.98-1.91) | 35.05-2.29 (2.38-2.29) | 36.70-3.26 (3.37-3.26) |
| No. of unique reflections | 25,300 (2,301) | 25,156 (2,171) | 14,789 (181) |
| Reflections in free set | 1254 (117) | 1301 (113) | 1431 (30) |
| $R_{work}$ | 0.18 (0.30) | 0.22 (0.26) | 0.26 (0.30) |
| $R_{free}$ | 0.22 (0.31) | 0.26 (0.31) | 0.30 (0.39) |
| NCS copies | 2 | 4 | 6 |
| No. of nonhydrogen atoms | | | |
| protein and zinc | 2,240 | 4,349 | 5,052 |
| solvent | 112 | 52 | 0 |
| Average B-factor (Å$^2$) | | | |
| protein and zinc | 63 | 58.9 | 72.77 |
| solvent | 53 | 53.1 | |
| Coordinate deviations | | | |
| bond lengths, Å | 0.019 | 0.004 | 0.005 |
| bond angles,° | 1.644 | 0.724 | 0.375 |
| *Validation and Deposition* | | | |
| Ramachandran plot | | | |
| favored,% | 99 | 99 | 97.4 |
| outliers,% | 0 | 0 | 0 |

*Table 1 continued on next page*

*Table 1 continued*

|  | Dimer | Dimer | Trimer |
|---|---|---|---|
| MolProbity clashscore | 3.73 | 2.00 | 2.31 |
| PDB ID | 5EIU | 5F7T | 5IEA |

Values in parenthesis are for the highest resolution shell.

miniTRIM constructs are excellent structural surrogates for the B-box/coiled-coil core of the full RBCC motif.

## Structures of the Bcc miniTRIM trimer and dimer

The Bcc miniTRIM trimer is organized as a triskelion: three B-box 2 domains make a central three-fold symmetric vertex from which the coiled-coil domains emanate as spokes (*Figure 3A*). As expected, the trimerization interactions are principally mediated by the B-box domain, burying 578 $\text{Å}^2$ of the available surface area from each subunit. This result indicates that the three-fold vertexes of the TRIM5α hexagonal lattice are made by B-box trimers. The B-box packs against the N-terminal end of the coiled-coil helix through a hydrophobic interface (*Figure 3A*, asterisks), suggesting that trimer formation requires the presence of the coiled-coil. This observation is consistent with the idea that higher-order assembly into a hexagonal lattice is a function of the integrated tripartite motif of TRIM5α, and does not simply arise from combining otherwise independent self-association motifs.

Interestingly, the majority of Bcc miniTRIM crystals we obtained (including three other forms that we did not refine) were composed of dimers (*Figure 3G* and *Supplementary file 1A*). The dimer is quasi-equivalent to the trimer: the same sets of residues make the same interactions but with valence of two instead of three. The dimer buries a smaller surface area per subunit (450 $\text{Å}^2$) compared to the trimer.

Both the trimer and dimer are stabilized by three layers of interactions, with a hydrophobic layer sandwiched between two hydrophilic layers (*Figure 3B,H*). The top layers (closest to the N-termini of the B-boxes) make a ring of salt bridges mediated by Glu102 and Lys103 from each subunit (*Figure 3D,J*). The middle, hydrophobic layers center on Trp117 (*Figure 3E,K*). In both oligomers, these indole sidechains make a hydrophobic core surrounded by a collar of close packed leucine sidechains (Leu105, Leu106, Leu118, and Leu132). The third, bottom layers also consist of a ring of salt bridges, this time with Glu120 and Arg121 (*Figure 3F,L*). At the periphery of this layer, the sidechain hydroxyl of Thr30 donates a hydrogen bond to Glu120. Interactions between the layers consist of a hydrogen bond between Glu102 (top) and the indole amide of Trp117 (middle), and close packing between the Trp117 sidechain (middle) and the Arg121 sidechain guanidinium group (bottom) (*Figure 3C,I*). The Trp117 indoles in the trimer form a channel that would be expected to have a highly negatively charged hole at the center (*Figure 3E*), which is likely to be partially stabilized by the positively charged Arg121 sidechains.

Comparison of the trimer and dimer bonding interactions revealed that the two oligomers are distinguished primarily by intermolecular packing between the Trp117, Leu118, and Leu132 sidechains in the hydrophobic layer. In particular, Leu118 and Leu132, which are located at the outer edges of the binding surface, are in van der Waals contact in the trimer, but more separated in the dimer (*Figure 3—figure supplement 1*). Another distinction is that the first two turns of the coiled-coil helix are overwound and bent in the dimer form (*Figure 3—figure supplement 2*), which might indicate some structural communication between the interacting B-boxes and the coiled-coil domain.

The apparent propensity of the B-box 2 domain to dimerize in our crystals prompted us to more closely analyze the oligomerization behavior of the miniTRIMs in solution. In order to avoid possible contributions from proximity-induced RING/RING interactions, we focused on the Bcc miniTRIM construct. We first analyzed Bcc miniTRIM by using sedimentation equilibrium analytical ultracentrifugation (AUC) at three different loading concentrations and three rotor speeds, and the combined data set was fit globally. In control experiments, both the W117E and R121E Bcc miniTRIM mutants were monomeric, in agreement with the structures (not shown), whereas the wildtype (WT) Bcc distributions indicated self-association (*Figure 3—figure supplement 3*). Global fitting of the WT

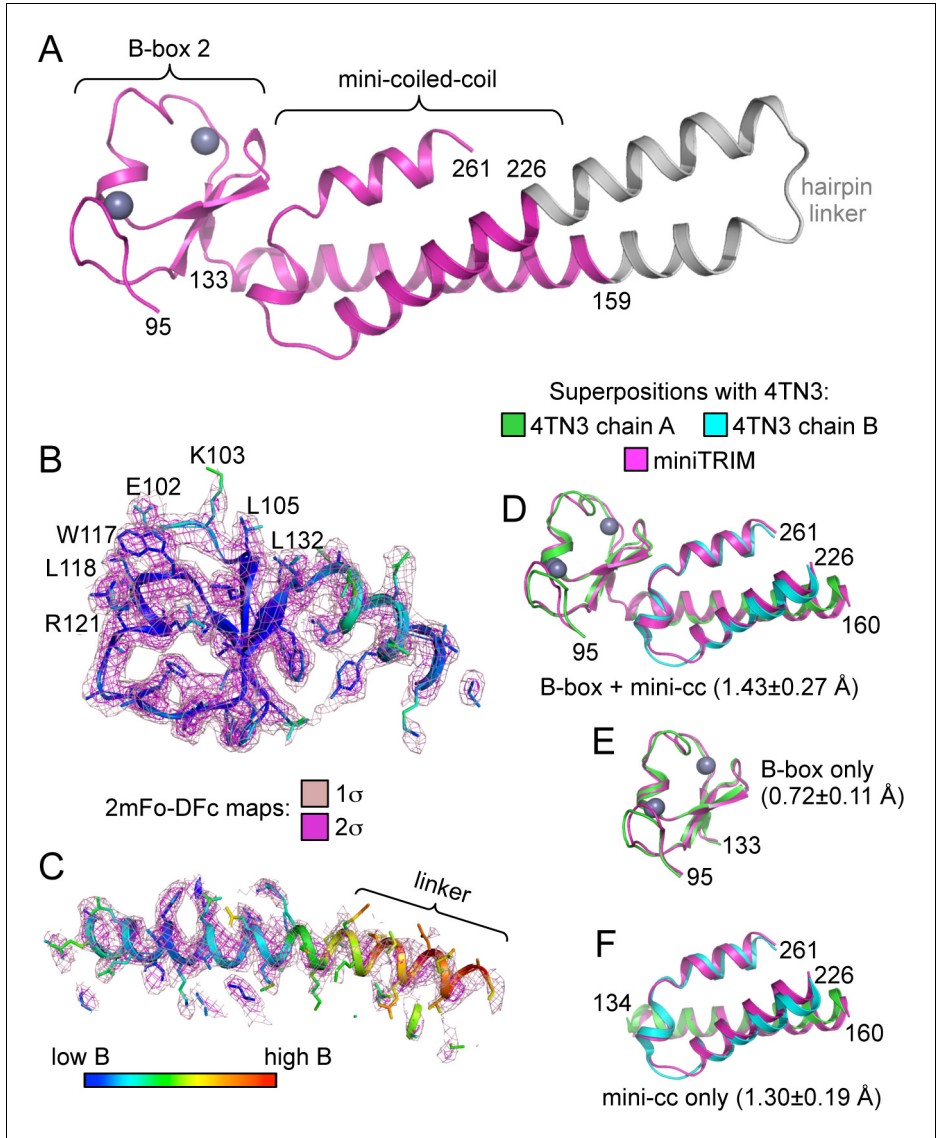

**Figure 2.** Structure of the Bcc miniTRIM. (**A**) Complete structure of the Bcc miniTRIM, from a dimer subunit. Residues derived from TRIM5α are colored in magenta, and the artificial hairpin linker is in gray. Charcoal gray spheres indicate zinc atoms. Residue numbers are indicated. (**B–C**) Electron density maps at two contour levels for (**B**) the B-box 2 domain and (**C**) the coiled-coil region of a trimeric Bcc subunit. The model is colored according to B-factor, which indicates that the B-box and proximal coiled-coil regions are well defined. B-box 2 sidechains within oligomerization interfaces are labeled to illustrate that these residues are well defined by the density. (**D–F**) Superposition of Bcc miniTRIM (magenta) with the corresponding B-box 2 and coiled-coil regions in the crystal structure of the rhesus TRIM5α B-box/coiled-coil fragment (PDB 4TN3) (**Goldstone et al., 2014**): (**D**) B-box and coiled-coil regions, (**E**) B-box alone, (**F**) coiled-coil alone. Residue ranges used in the superposition are indicated, as are the average mean square deviations ± s.d. from pair-wise superpositions of each of the 12 monomer structures. Deviations from each individual superposition are in **Supplementary file 1B**.

The following figure supplement is available for figure 2:

**Figure supplement 1.** Ribbon representations of the 12 crystallographically independent Bcc miniTRIM structures solved in this study.

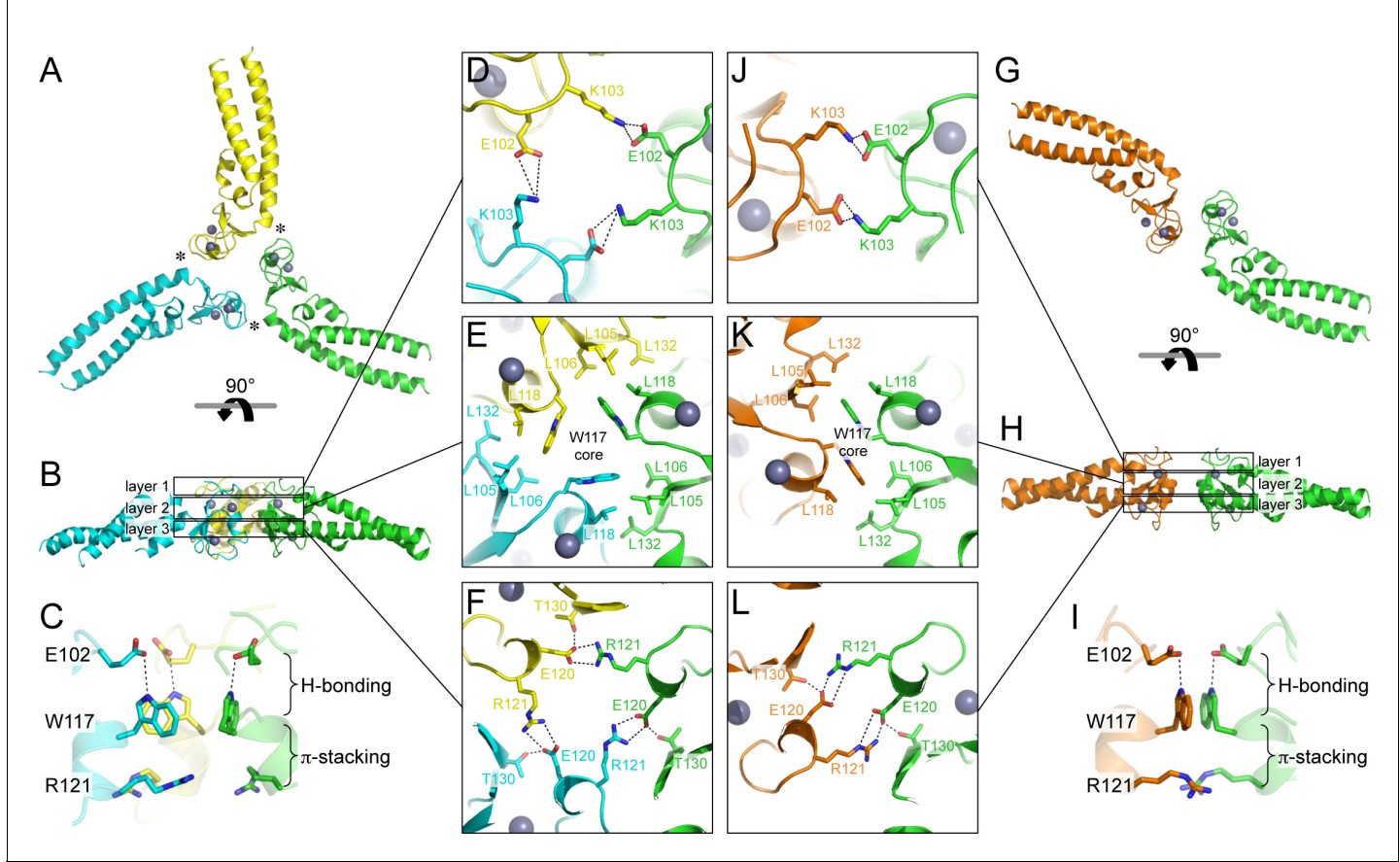

**Figure 3.** Oligomeric structures of Bcc miniTRIM. (**A**) Trimer crystal structure and (**G**) dimer crystal structure, viewed from the 'top' (closest to the N-termini). (**B,H**) Side views. Three layers of interactions are boxed and expanded in the central panels. Asterisks in A indicate a site of close packing between the B-box and the N-terminal end of the coiled-coil helix. (**C,I**) Interactions between layers. (**D–F,J–L**) Expanded views of three layers of interactions, in the same orientations as A and G. Relevant sidechains are shown as sticks and labeled. Hydrogen bonds and salt bridges are indicated by black dashed lines. Zinc atoms are shown as gray spheres.

The following figure supplements are available for figure 3:

**Figure supplement 1.** Comparison of the B-box trimer (**A**) and dimer (**B**).

**Figure supplement 2.** Superposition of representative subunits from the trimer (cyan) and dimer (green) indicate local bending of the coiled-coil helix spanning residues 133–139.

**Figure supplement 3.** Sedimentation equilibrium analytical ultracentrifugation profiles of Bcc miniTRIM.

**Figure supplement 4.** SEC-MALS analysis of Bcc miniTRIM.

**Figure supplement 5.** Slight clam shell-like opening of the B-box dimer interface.

equilibrium distributions to a monomer-dimer-trimer model only returned stable values for the monomer-dimer dissociation constant (1 µM). Indeed, the AUC data gave a satisfactory fit to a monomer-dimer equilibrium model, indicating that trimers (or higher-order oligomers than dimers) were disfavored in solution. To determine whether higher-order oligomers would form at higher protein concentrations, we also analyzed the Bcc miniTRIM at 0.9 mM by using analytical size exclusion chromatography with multi-angle light scattering (SEC-MALS). The mass trace (blue curve in *Figure 3—figure supplement 4*) had a sharp peak at the leading edge of the protein absorbance peak (black curve), a curved plateau in the central region above the expected mass for a dimer, and

tapered towards the expected monomer mass at the tail. This again indicated dynamic equilibrium between monomer and dimer states, and that higher-order species were also present in the main peak, but at a much smaller fraction than dimers. Thus, the Bcc miniTRIM was predominantly dimeric in solution.

The solution behavior of Bcc miniTRIM is in sharp contrast to that of full-length protein. Published AUC analysis of freshly purified full-length TRIM5 did not show evidence of B-box mediated self-association, even at a loading concentration that is six-fold higher than our measured miniTRIM dimerization affinity (*Langelier et al., 2008*). Uncoupling of the B-box from the full coiled-coil and RING domains therefore appears to have amplified its propensity for oligomerization. Another difference is that, upon incubation, full-length TRIM5α has a clear propensity to assemble into hexagonal arrays – and by inference, three-way B-box/B-box interactions – even at low µM concentrations (*Ganser-Pornillos et al., 2011*; *Li et al., 2016*; and this study). Since the miniTRIMs lack an intact coiled-coil, extended incubation did not result in lattice formation but only produced higher molecular weight aggregates (not shown). Our interpretation of all of these observations is that artificial uncoupling of the B-box 2 domain from the full coiled-coil and RING domains has stabilized the B-box dimer form. In context of the full-length protein, assembly cooperativity drives the B-box into the trimer form, and so the B-box dimer might represent an intermediate state that can incorporate an additional B-box domain to form the trimer. Consistent with this interpretation, our structures indicate that the Bcc miniTRIM dimers can open slightly like a clam shell, which we imagine can lead to a wider opening and provide access to a third subunit (*Figure 3—figure supplement 5*). We note that this interpretation does not preclude a functional role for a B-box dimer. For example, it is possible that B-box dimerization might be a mechanism to promote RING dimerization and E3 ligase activation. Nevertheless, it seems clear that the B-box trimer is what facilitates hexagonal lattice assembly, at least in vitro.

## Structure-based TRIM5α B-box 2 mutants are impaired in restriction and capsid binding

To test the functional relevance of the B-box mediated interactions in our structures, we generated structure-based mutations in full-length rhesus TRIM5α and tested the mutant proteins for their ability to inhibit transduction of GFP-labeled HIV-1 in HeLa cells (*Figure 4*, *Table 2*). Charge reversal mutations in the top layer of interactions (E102K, K103E, E102K/K103E) diminished restriction activity (*Figure 4A*), confirming that this ring of salt bridges has an appreciable contribution to higher-order TRIM5α assembly. Individual charge reversals in the bottom layer (E120R, R121E) severely abrogated restriction activity (*Figure 4B*), in agreement with previous work (*Diaz-Griffero et al., 2009*; *Li and Sodroski, 2008*). However, the E120R/R121E double mutation did not have a compensatory effect. This is in contrast with previous studies (*Diaz-Griffero et al., 2009*; *Li and Sodroski, 2008*) but is consistent with the structures since Arg121 is involved in both a salt bridge and a hydrophobic packing interaction with Trp117. The repositioned guanidinium group in E120R seems unlikely to generate the optimal geometry for these interactions.

In the middle, hydrophobic layer, the W117E mutation was previously shown to abolish restriction activity (*Diaz-Griffero et al., 2009*), which we have confirmed here (*Figure 4C*). Interestingly, however, the W117A mutant still harbored some restriction activity, most likely because the hydrophobic layer remains stabilized by an outer ring of leucines. As with Trp117, alanine substitution mutants for Leu118 (*Figure 4D*) and Leu132 (*Figure 4E*) had measurable restriction activity, whereas aspartate substitutions were more significantly disruptive. Of the remaining two leucines, L105A had full activity and L106A did not restrict HIV-1 (*Figure 4F*). These results are again consistent with the structures since Leu105 is only partially buried, whereas Leu106 is completely buried within both the dimer and trimer interfaces. Interestingly, we found that steady state expression levels of the TRIM5α B-box 2 domain mutants varied considerably, as noted previously (*Diaz-Griffero et al., 2007*), and that furthermore there was an inverse correlation between the steady state expression levels of the mutant proteins and restriction activity (*Figure 4G*). The best expressing mutants (e.g., W117E and R121E) did not restrict HIV, whereas the lowest expressing mutant (L105A) had WT-like restriction activity. The expression levels also inversely correlated with in vitro assembly efficiency – W117E and R121E did not assemble, whereas L105A assembled efficiently (*Ganser-Pornillos et al., 2011*; also see below). We therefore speculate that the ability to assemble reduces steady state protein levels because TRIM5α proteins that assemble are turned over more rapidly in cells. An

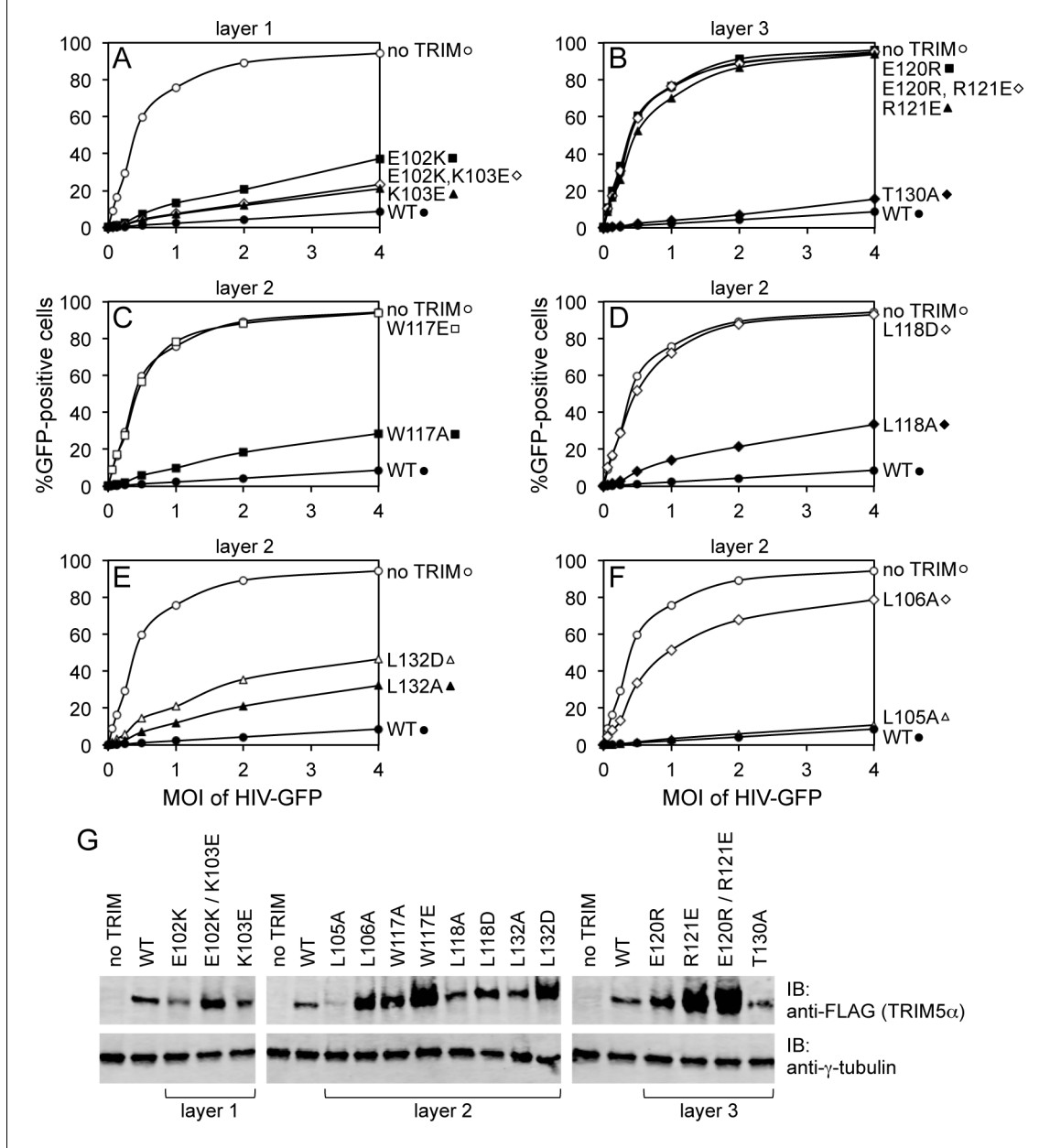

**Figure 4.** Restriction activities of rhesus TRIM5α containing structure-based B-box 2 domain mutations. For clarity, the data are presented in multiple panels. (A–F) GFP-labeled HIV-1 efficiently transduced HeLa cells that did not express exogenous TRIM5α (no TRIM, open spheres). Transduction was significantly inhibited in cells that expressed WT TRIM5α (WT, filled spheres). The same results were obtained in two independent experiments. (G) Immunoblots (IB) of whole cell lysates with anti-FLAG antibody to determine expression levels of rhesus TRIM5α mutants (upper). Anti-tubulin blots indicate that equivalent samples were loaded into each lane (lower).

extension of this argument is that the ability of TRIM5α to assemble correlates with its ability to restrict HIV-1.

We also tested the effect of the hydrophobic B-box 2 domain mutations on capsid binding activity in vitro. For these experiments, we used TRIM5-21R, a chimeric construct described in previous studies as a functional and structural surrogate for TRIM5α due to its more favorable biochemical properties (*Diaz-Griffero et al., 2006a*; *Ganser-Pornillos et al., 2011*; *Kar et al., 2008*; *Langelier et al., 2008*). We incubated crosslinked HIV-1 CA tubes (biochemical surrogates for the HIV-1 capsid) with WT and mutant TRIM5-21R proteins, and then measured binding in a co-pelleting

**Table 2.** Summary of structure-based mutagenesis.

| Mutation | Structure context | Restriction activity | Capsid binding | Spontaneous assembly | Co-assembly | Templated assembly |
|---|---|---|---|---|---|---|
| None (WT) | | ++ | ++ | hexagonal[1] | ++ | ++ |
| E102A | top layer | + | n.d. | n.d. | n.d. | n.d. |
| K103A | top layer | + | n.d. | n.d. | n.d. | n.d. |
| E102A,K103A | top layer | + | n.d. | n.d. | n.d. | n.d. |
| L105A | middle layer | ++ | ++ | macramé[1] | ++ | − |
| L106A | middle layer | − | + | −[1,2] | − | n.d. |
| W117A | middle layer | + | + | −[1,2], striated[3] | + | − |
| W117E | middle layer | − | + | −[1,2] | − | n.d. |
| L118A | middle layer | + | ++ | −[1], hexagonal[2] | n.d. | n.d. |
| L118D | middle layer | − | − | −[1,2] | − | n.d. |
| L132A | middle layer | + | + | −[1], hexagonal[2] | n.d. | n.d. |
| L132D | middle layer | +/− | + | −[1,2] | − | n.d. |
| E120R | bottom layer | − | n.d. | n.d. | n.d. | n.d. |
| R121E | bottom layer | − | +* | n.d. | n.d. | n.d. |
| E120R,R121E | bottom layer | − | n.d. | n.d. | n.d. | n.d. |
| T130A | bottom layer | ++ | n.d. | n.d. | n.d. | n.d. |

[1]At 1 mg/mL in standard low salt buffer
[2]At >5 mg/mL in standard low salt buffer
[3]At >5 mg/mL in high salt buffer
*from previous study (**Ganser-Pornillos et al., 2011**)
n.d. – not determined

assay (**Fribourgh et al., 2014**; **Ganser-Pornillos et al., 2011**; **Stremlau et al., 2006**). In control experiments, about 50% of WT TRIM5-21R was consistently found in the pellet (**Figure 5A**). Mutants were analyzed in parallel with a WT control, and results in **Figure 5B** are representative of mutant binding efficiencies relative to WT from at least two experiments with independent protein preparations. Essentially all of the B-box mutants tested had detectable capsid binding activity, which was expected because all of these constructs contained an intact SPRY domain and because the R121E mutation that abolished restriction still supported capsid binding in a similar assay (**Ganser-Pornillos et al., 2011**). Nevertheless, our results showed that weaker or no capsid binding correlated with weaker or no restriction activity (**Table 2**), confirming that B-box 2 domain mediated interactions promote efficient capsid recognition.

## Assembly properties of TRIM5α B-box 2 domain mutants

We then directly determined the effects of the B-box mutations on TRIM5α assembly activity in vitro (**Figures 6–8**, and **Table 2**). For this analysis, we focused on the hydrophobic (layer 2) mutations. Previously, purified TRIM5-21R was shown to assemble spontaneously into hexagonal arrays when incubated in low salt buffer at about 1 mg/mL (**Ganser-Pornillos et al., 2011**). These micron-sized arrays can be readily visualized by negative stain electron microscopy (**Figure 6A**). In contrast, the R121E mutant did not assemble spontaneously even at concentrations up to 30 mg/mL (**Ganser-Pornillos et al., 2011**). We therefore tested our TRIM5-21R mutants for spontaneous assembly at both low (1 mg/mL) and high (>5 mg/mL) protein concentrations. We found that mutations that gave the most significant reductions in restriction activity (L106A, W117E, L118D, and L132D) also prevented assembly at all protein concentrations tested (up to 18 mg/mL) (**Table 2** and data not shown). Two of the mutations that supported intermediate restriction activity (L118A and L132A) also supported hexagonal lattice assembly, but only at high protein concentrations and to a more limited extent compared to WT (**Figure 6B,C**). A third intermediate mutation, W117A, altered the

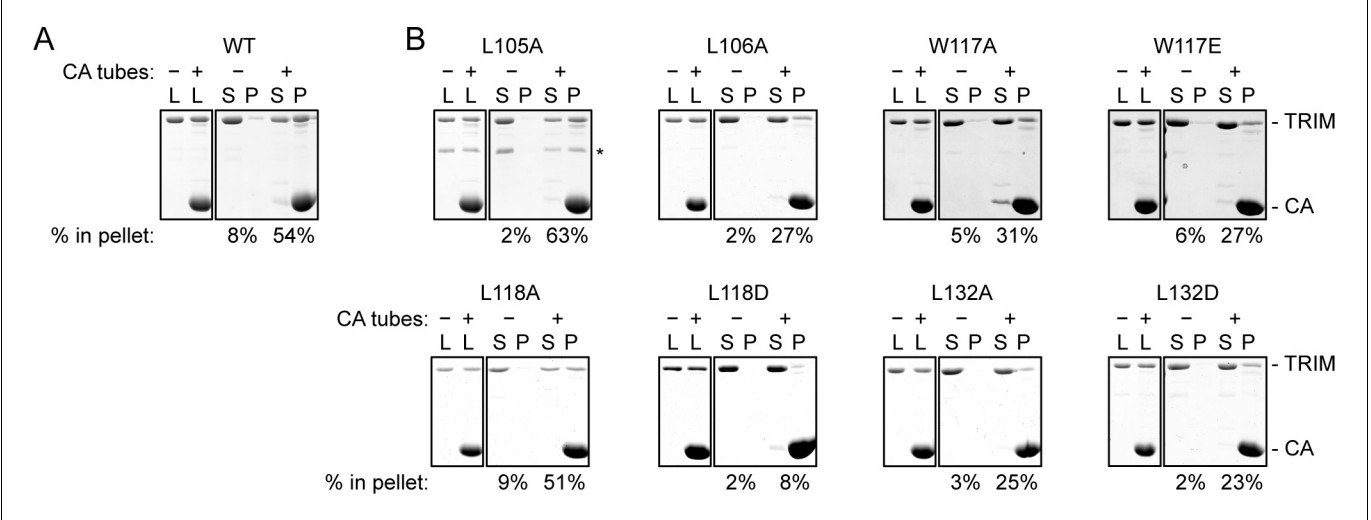

**Figure 5.** CA tube pull-down assay. (**A**) WT control. (**B**) Indicated mutants. Purified TRIM5-21R proteins were incubated with disulfide-stabilized CA tubes and pelleted in a microcentrifuge. Bound (pellet) and unbound (supernatant) proteins were visualized by SDS-PAGE with Coomassie staining and quantified. Percentage values indicate the fraction of protein in the pellet. Results are representative of at least two experiments per mutant, each done with an independent protein preparation. L, load; S, supernatant; P, pellet. CA and TRIM bands are indicated. The asterisk indicates an apparent proteolytic fragment of TRIM5-21R.

assembly phenotype of TRIM5-21R. This mutant failed to assemble in standard low salt buffer at both 1 and 5 mg/mL, but at high protein concentrations and higher salt (>250 mM NaCl), it assembled into striated arrays (*Figure 6D*). Interestingly, the L105A mutation (full restriction activity) also altered the spontaneous assembly behavior of TRIM5-21R. Under the same conditions as WT (low salt buffer and 1 mg/mL protein), L105A formed macramé-like networks that appeared distinct from either the hexagonal or striated arrays (*Figure 6E*).

Two-dimensional crystals of HIV-1 CA-NC, which mimic the hexagonal HIV capsid lattice at its planar limit, can promote assembly of TRIM5-21R and native TRIM5α proteins into flat hexagonal arrays (*Ganser-Pornillos et al., 2011*; *Li et al., 2016*). We therefore tested whether the CA-NC arrays can 'rescue' the altered spontaneous assembly phenotype of L105A and W117A. However, results were inconclusive; although the TRIM proteins appeared to associate with the CA-NC arrays, diffraction patterns that would indicate overlaying lattices were not observed (*Table 2* and data not shown). We therefore used an alternative assay wherein soluble TRIM5α and HIV-1 CA are mixed and incubated under basic conditions and moderate salt concentrations, which results in formation of CA tubes that are almost uniformly decorated with TRIM (*Li et al., 2016*). The overlaying TRIM lattice on these tubes is related to the flat hexagonal arrays, except that it now follows the basal curvature of the CA tubes. Both WT TRIM5-21R (*Figure 7A*) and the L105A mutant (*Figure 7B*) showed similar behavior in this assay; that is, virtually all the tubes were uniformly decorated, and the decorations looked similar to each other and to those made by WT TRIM5α (*Li et al., 2016*). In contrast to the flat arrays, hexagon shapes were more difficult to discern in projection images of the curved arrays, and so we also analyzed negatively stained samples by electron tomography. As observed with vitrified samples (*Li et al., 2016*), densities surrounding the CA tubes were readily discernible in the tomograms of our negatively-stained samples (*Figure 8*). As expected, the peripheral TRIM5α layers exhibited a high degree of disorder, but for both WT (*Figure 8A*) and L105A (*Figure 8B*), we could clearly observe regions of local order with hexagonal rings having the expected dimensions for a TRIM hexamer. Interestingly, a proportion of the L105A decorations appeared more ladder-like (*Figure 8C*), but again the distances between the repeating units were similar to the TRIM hexagon dimensions. We therefore conclude that, although there is no clear structural explanation for the spontaneous assembly behavior of the L105A mutant, it can nevertheless form hexagonal arrays when it binds CA tubes in vitro. The inability of this mutant to form flat lattices (the endpoints of

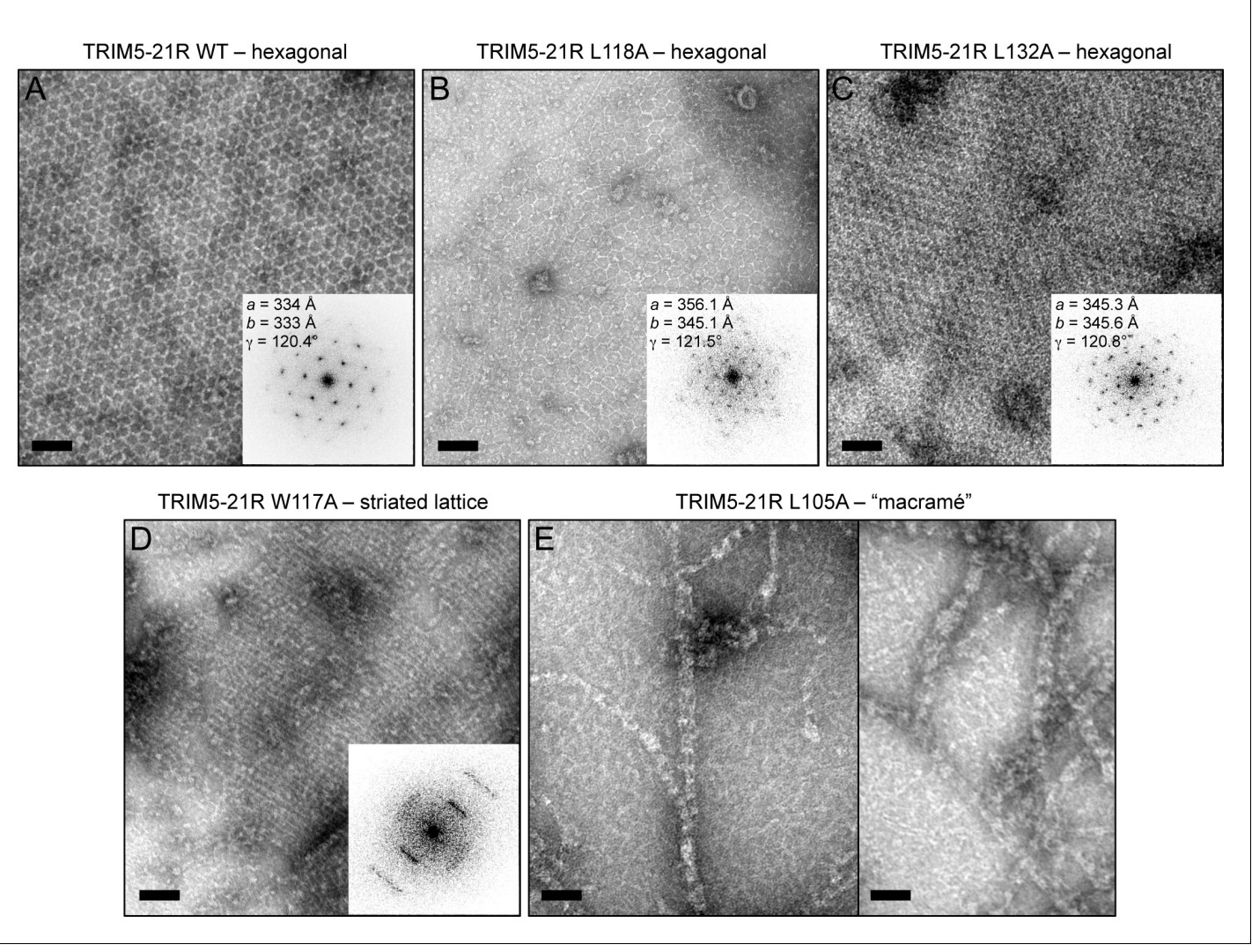

**Figure 6.** Spontaneous assembly of TRIM5-21R. (**A**) WT TRIM5-21R spontaneously assembled into hexagonal arrays at a concentration of 1 mg/mL in 25 mM Tris, pH 8, 25 mM NaCl, 1 mM TCEP (standard conditions) (*Ganser-Pornillos et al., 2011*). Main panel shows a representative negatively stained image of the arrays; inset shows a Fourier transform of the image. The unit cell spacing (symmetry unimposed) calculated from the diffraction pattern is indicated. (**B**) L118A and (**C**) L132A, which gave intermediate restriction phenotypes in context of rhesus TRIM5α, also assembled into hexagonal nets, but at higher protein concentrations (2 and 9 mg/mL, respectively). (**D**) W117A aggregated under standard conditions but at >5 mg/mL and 250 mM NaCl assembled into a striated array. (**E**) L105A, which was fully restriction competent, assembled spontaneously under standard conditions into networks that were neither hexagonal not striated. Results are representative of two or three experiments per construct, each done with an independent protein preparation. Scale bars = 100 nm.

both the spontaneous assembly and template driven methods) suggests that B-box/B-box interactions might have some link to lattice curvature.

We also used the co-assembly assay to characterize the W117A mutant that assembled into striated arrays (*Figure 6D*). This mutant also decorated the CA tubes, but the decorations were more limited in extent compared to either WT or L105A, and regions of undecorated CA were more readily apparent in the projection images (*Figure 7C*). Similarly, the tomograms revealed considerable disorder in the overlaying TRIM lattice, and hexagon-shaped decorations were not easily discerned (not shown).

In summary, our analysis of structure-based B-box 2 domain mutations support a general correlation between TRIM5α restriction activity in cells, capsid binding efficiency in vitro, and hexagonal lattice assembly in vitro. Our results therefore fit the model wherein both capsid recognition and

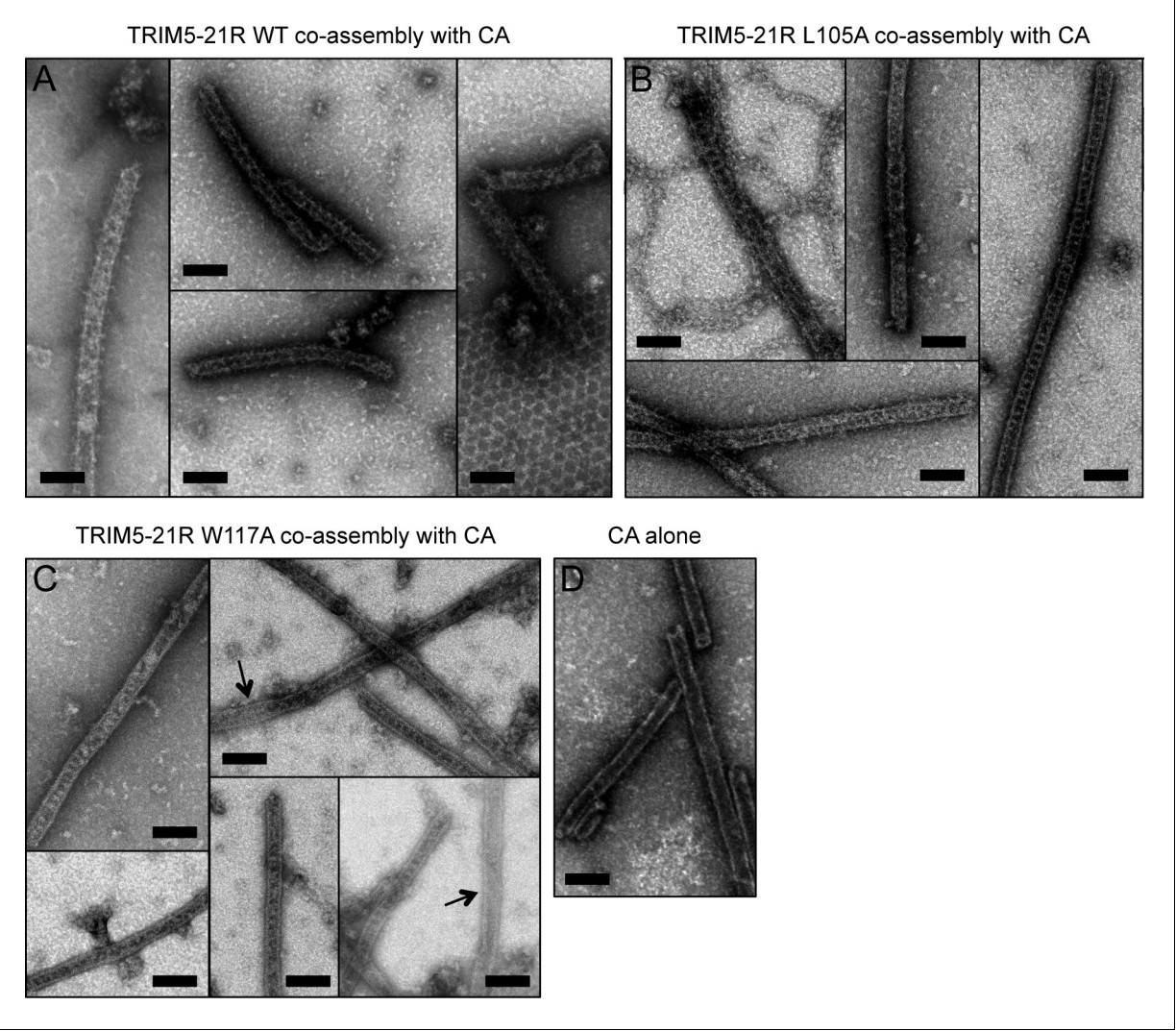

TRIM5-21R WT co-assembly with CA

TRIM5-21R L105A co-assembly with CA

TRIM5-21R W117A co-assembly with CA

CA alone

**Figure 7.** Co-assembly of TRIM5-21R with HIV-1 CA. (**A**) Incubation of WT TRIM5-21R with soluble HIV-1 CA protein induced assembly of TRIM-coated capsid tubes. A similar phenotype was observed when co-assembly is performed with African green monkey TRIM5α (*Li et al., 2016*). (**B**) TRIM5-21R with the L105A mutation made similar decorated tubes as WT in this assay. (**C**) W117A also made similar decorations, but to a more limited extent. Partially decorated and undecorated tubes were more prevalent (arrows). Results are representative of two or three experiments per construct, each done with an independent protein preparation. (**D**) Undecorated CA tubes shown for comparison. Scale bars = 100 nm.

antiviral restriction are facilitated by B-box mediated assembly of TRIM5α into a hexagonal lattice that wraps around the viral capsid.

## Molecular modeling of the TRIM5 hexagonal lattice

The low-resolution structure of the TRIM5α hexagonal lattice consists of large lobes of density at the two-fold and three-fold symmetric positions connected by thin linkers of density (*Ganser-Pornillos et al., 2011*; *Li et al., 2016*). The thin linkers are made by the antiparallel coiled-coil dimer scaffold and the two-fold lobes are made by the SPRY domain (*Ganser-Pornillos et al., 2011*; *Goldstone et al., 2014*; *Li et al., 2016*; *Sanchez et al., 2014*; *Weinert et al., 2015*; *Li et al., 2016*). Our structural and biochemical analyses indicate that the three-fold vertexes are made by B-box 2 domain trimers. By combining our miniTRIM trimer crystal structure with that of the B-box/coiled-coil dimer (PDB 4TN3) (*Goldstone et al., 2014*), we built a molecular model of a flat TRIM5α hexagonal lattice (*Figure 9A*). The unit cell spacing of this model is 338.5 Å, which is an almost exact match to the observed value for rhesus TRIM5α (340 Å) (*Li et al., 2016*). All of the SPRY domains

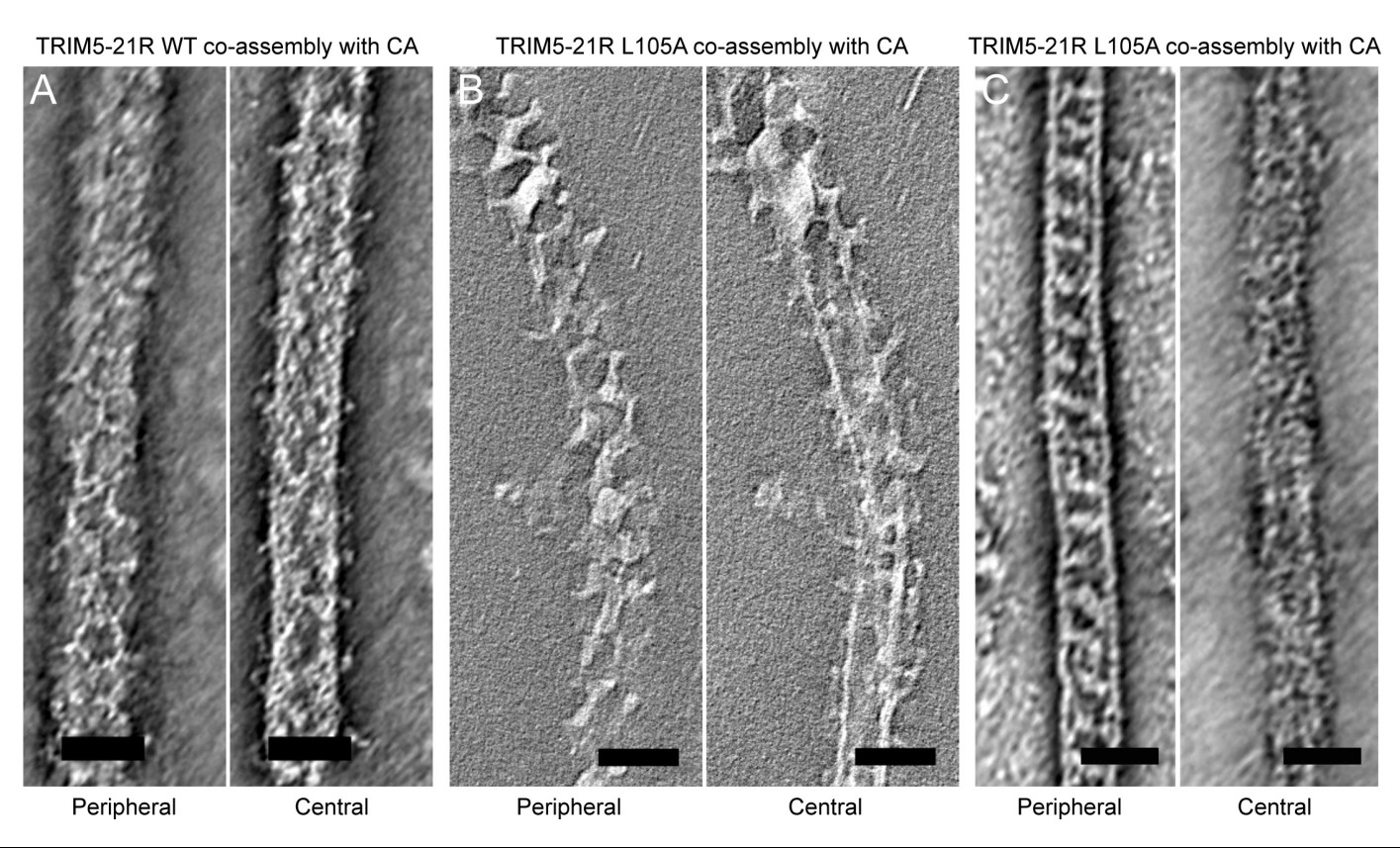

TRIM5-21R WT co-assembly with CA    TRIM5-21R L105A co-assembly with CA    TRIM5-21R L105A co-assembly with CA

Peripheral    Central        Peripheral    Central        Peripheral    Central

**Figure 8.** Slices of tomographic reconstructions of (**A**) WT and (**B**) L105A TRIM5-21R coated CA tubes. Left panels show peripheral slices, and right panels show central slices of the same tube. (**C**) Some of the L105A-coated tubes had ladder-like TRIM overlays that do not seem hexagonal. These could be due to overlapping lattices or an alternative arrangement of TRIM dimers. Scale bars = 50 nm.

are located on one side of the lattice plane (*Figure 9B*, blue spheres), where they could bind the capsid simultaneously. At the three-fold vertexes, the N-terminal ends of the B-boxes point toward the other side of the lattice plane. This suggests that the RING domains are oriented away from the capsid (*Figure 9B*, red spheres). This molecular architecture indicates that the higher-order TRIM scaffold also compartmentalizes the biochemical activities of the RING and SPRY domains.

## The B-box/coiled-coil interface allows rigid body motions of the two domains

When TRIM5α binds to a retroviral capsid, it must accommodate the variable surface curvature of the capsid. Capsid lattice curvature is generated by rigid body hinge motions between subunits, and we have previously identified such hinges by comparing crystallographically independent structures of the hexameric and pentameric capsid building blocks (*Pornillos et al., 2009*; *2011*). The availability of 12 independent miniTRIM structures allowed us to perform a similar analysis here. As illustrated in *Figure 10A*, superposition of the miniTRIM structures revealed that the coiled-coil can swing relative to the B-box 2 domain. Flexion occurs about the B-box/coiled-coil interface, which is lined almost exclusively by aliphatic sidechains (*Figure 10B*). Note that in context of full-length TRIM5, this 'greasy' interface is a quaternary contact between the subunits of the native dimer. Bending of the first two turns of the coiled-coil helix in the dimer subunits produced the greatest change in orientation of the B-box relative to the coiled-coil, but the rigid body motions were also evident in comparing dimer subunits or trimer subunits alone (not shown). In context of the trimer, the three coiled-coils do not contact each other and can therefore move independently (*Figure 10C,*

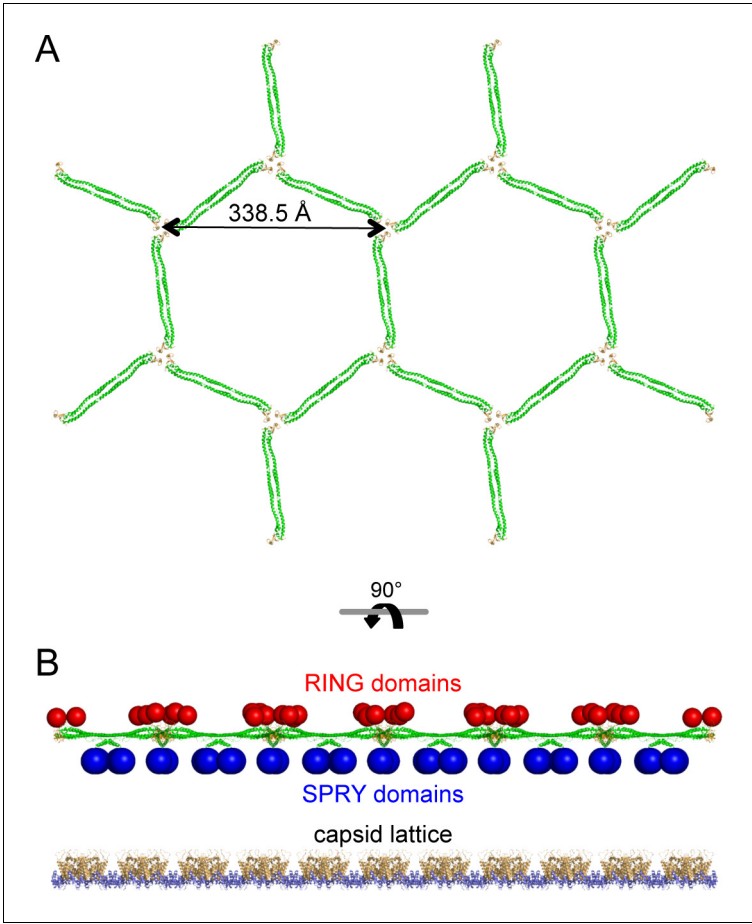

**Figure 9.** Molecular model of a flat TRIM5α hexagonal lattice. (**A**) Top view, with the B-box 2 domains colored in orange and the coiled-coil domains in green. (**B**) Side view, with the expected positions of the SPRY domains (blue) and RING domains (red) indicated by spheres. The flat capsid lattice is shown for reference.

**D**). We conclude that TRIM5α uses the same general mechanism as retroviral capsids – flexion across quaternary interfaces and local conformational variations – to generate variable lattice curvature.

## B-box/B-box interactions stereochemically restrict RING domain activation

In addition to promoting avid capsid binding, B-box mediated interactions are expected to promote clustering of the upstream RING domain. Indeed, TRIM5α assembly on retroviral capsids is reported to enhance E3 ligase activity (*Pertel et al., 2011*). The RING domain dimerizes to bind E2-Ub conjugates and catalyze Ub transfer (*Yudina et al., 2015*). So far, we have been unable to solve a crystal structure of the RBcc miniTRIM, but structures of monomeric (inactive) and dimeric (active) forms of the TRIM5α RING domain are both known (*Lienlaf et al., 2011*; *Yudina et al., 2015*), and so we used molecular modeling to determine possible configurations of the RINGs relative to the B-box trimer. An important consideration here is the structure of the L1 linker that connects the RING and B-box domains, which we define as the 23 amino acids (residues 72–94) that link the globular zinc-coordinating folds. Residues 72–82 are disordered in the monomeric RING structure, but are folded into a 4-helix bundle in the dimeric RING (*Lienlaf et al., 2011*; *Yudina et al., 2015*). Thus, a RING monomer has a longer (and presumably more flexible) linker to the B-box than a RING dimer subunit.

We used the program RANCH (*Bernadó et al., 2007*) to calculate an ensemble of 10,000 models wherein the monomeric RING is flexibly tethered to the B-box trimer. Linker residues were modeled as impenetrable spheres, and the entire linker was assumed to be an 'intrinsically disordered'

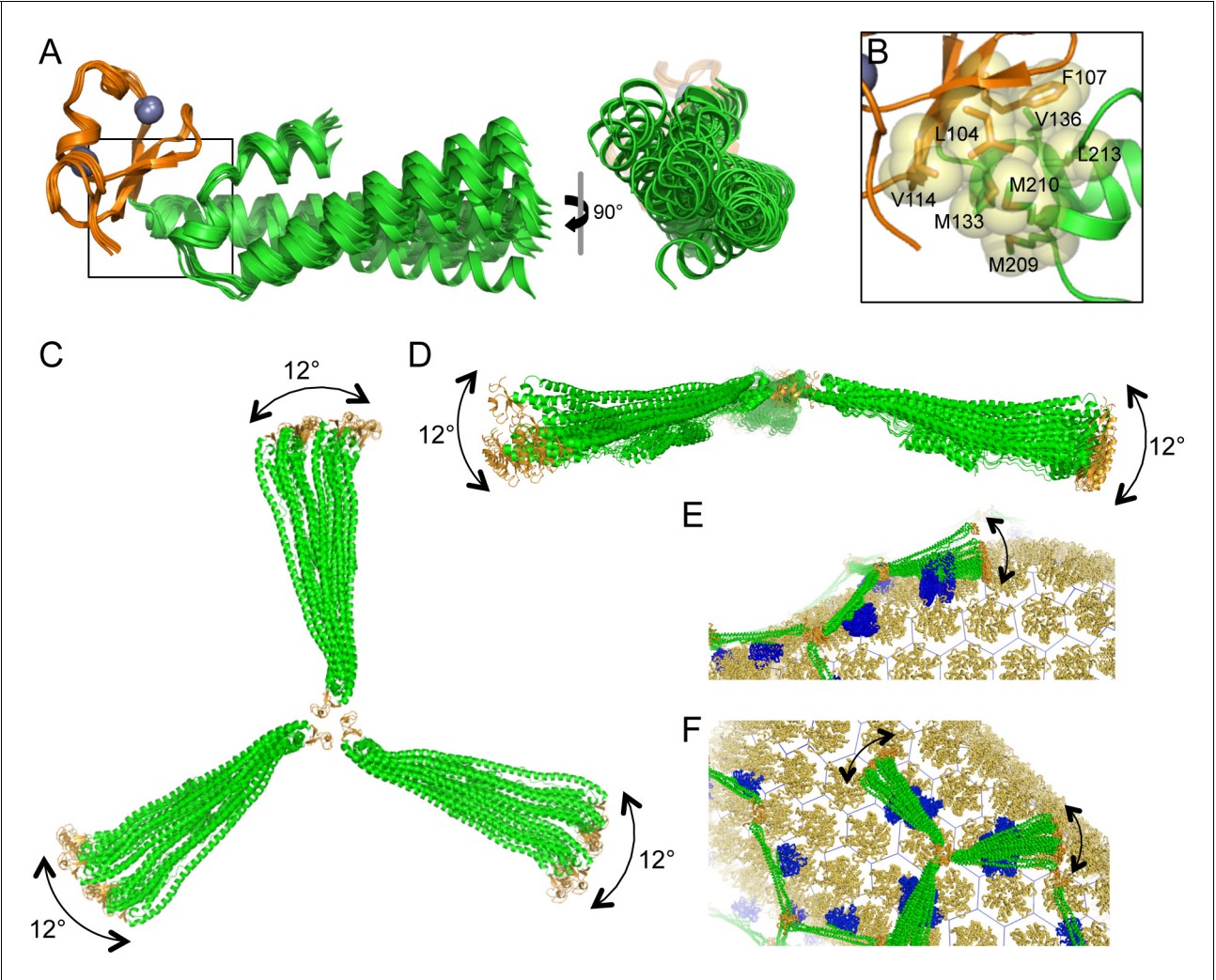

**Figure 10.** Flexible architecture of the miniTRIMs. (A) Orthogonal views of 12 crystallographically independent structures of Bcc miniTRIM. Superpositions of the structures on the B-box 2 domains (orange) reveals rigid body movements of the coiled-coil domains (green). (B) Close-up view of the B-box/coiled-coil interface boxed in A. Relevant sidechains are shown explicitly and labeled. (C,D) Superpositions of multiple full-length triskelion models on the B-boxes illustrate that the coiled-coil arms can swing flexibly relative to the B-box trimer vertex. (E,F) Speculative illustrations of how flexible triskelion arms can simultaneously allow the assembling TRIM lattice (green – coiled-coil; orange – B-box) to follow the curvature of the capsid (yellow orange) while scanning for optimal binding positions of the SPRY domains (blue).

segment ('native' setting in RANCH). This treatment seemed appropriate since the linker adopts alternative secondary structures (*Lienlaf et al., 2011*; *Yudina et al., 2015*). In the resulting ensemble, the vast majority of the RING domains are located above the plane of the trimer (*Figure 11A*, right). Thus, even flexibly tethered monomeric RINGs are predominantly located on one side of the lattice and away from the capsid. In a subset of our models, the RING/B-box linker folds down, in a configuration that brings the RING domain against the trimer vertex (*Figure 11A*, left). In principle, close packing of the RING domain and/or the RING/B-box linker against the trimer interface might explain the observation that the RING domain contributes to the efficiency of TRIM5α higher-order assembly (*Li et al., 2011*).

In the RING dimer, residues 72–82 fold into a 4-helix bundle, and destabilizing mutations in this region abrogate E3 ligase activity (*Yudina et al., 2015*). A RING dimer subunit is therefore separated from its B-box by a shorter linker of 12 residues ($_{83}$EVKLSPEEGQKV$_{94}$). Analysis of this sequence using the PEP-FOLD server (*Shen et al., 2014*) predicts that residues 87–94 have some propensity to fold into a short helix, and we therefore speculate that the dimeric RING/B-box linker

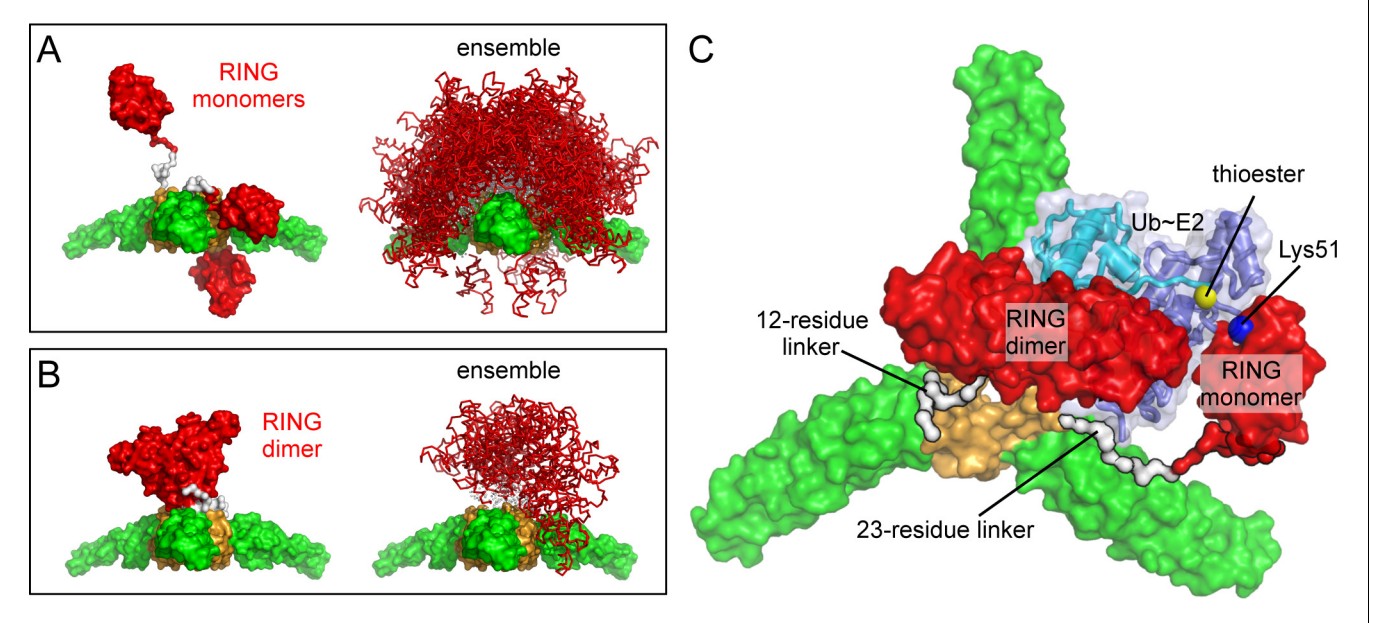

**Figure 11.** Models of RING domain configurations in context of the B-box trimer. (**A**) RING domains were modeled as monomers attached to their respective B-boxes by a flexible 23- residue linker. Left panel shows three representative configurations of the RING monomer relative to the trimer plane; above, within, and below. Right panel shows 500 of the 10,000 models calculated. (**B**) RING domains were modeled as dimers with a shorter, 12-residue linker. This resulted in only 13 configurations that were sterochemically plausible. (**C**) Computational model of a self-ubiquitination complex. Domains and proteins are color-coded as follows: RING, red; L1 linker, red (N-term) and white (C-term); B-box 2, orange; coiled-coil, green; E2, dark blue; ubiquitin, cyan. Positions of the thioester (yellow) and Lys51 amine (blue) are indicated by spheres, and are about 7 Å apart in this model.

might actually be a hinge rather than a flexible tether. Since E2-Ub binding imposes additional spatial constraints, stereochemical clashes can only be avoided if the RING dimer is positioned above the B-box trimer (*Figure 11B*). We therefore conclude that in context of the TRIM hexagonal lattice, the B-box trimer spatially restricts the RING domain, such that an active E3 ligase can only form on the side of the lattice that faces the cytoplasm. We also performed the same analysis on the B-box dimer; as expected, it also restricted RING positions, but to a lesser extent than the trimer (not shown).

The functional target of TRIM5α ubiquitination has not been determined definitively, but self-ubiquitination correlates with inhibition of retroviral reverse transcription (*Campbell et al., 2015*; *Fletcher et al., 2015*). The principal Ub attachment sites are in the RING domain (Lys45 and Lys51) (*Fletcher et al., 2015*). Although a B-box dimer is more naturally compatible with a RING dimer, a B-box trimer suggests an intuitively appealing mechanism for self-ubiquitination because it clusters three RING domains. In this model, two of the RING domains would dimerize and orient an E2-conjugated Ub for nucleophilic attack by the third RING (*Yudina et al., 2015*). To determine if such a mechanism of self-ubiquitination is stereochemically plausible, we constructed a model for a ubiquitination complex on a TRIM5α trimer vertex, by adding a third RING and an E2-Ub conjugate to one of our RING dimer/B-box trimer models. The two RING subunits that form the dimer were connected to their corresponding B-boxes by 12-residue linkers, whereas the third RING was connected by a longer and more flexible 23-residue linker. As illustrated in *Figure 11C*, we were able to identify stereochemically acceptable conformations that simultaneously allow trimerization of the B-box, dimerization of two RINGs, binding of the RING dimer to a Ub-conjugated E2, and positioning of the appropriate lysine in the third RING for nucleophilic attack of the E2-Ub thioester bond. Thus, although our modeling approach is somewhat crude, the results indicate that B-box trimerization is compatible with RING activation and TRIM5α self-ubiquitination.

## Discussion

### Hexagonal lattice model for avid capsid binding

The 'pattern recognition' model of capsid binding postulates that higher-order assembly of TRIM5α into a hexagonal lattice positions multiple SPRY domains to match both the orientations and spacing of their binding epitopes on the capsid surface (*Ganser-Pornillos et al., 2011*; *Li et al., 2016*). Our studies are consistent with this model, and further indicate that three-fold symmetric interactions at the vertexes of the hexagonal net are directly mediated by the B-box 2 domain. Indeed, our analysis suggests a 'rank order' of avidity promoting interactions, beginning with the 'minimal' coiled-coil dimer unit that positions (or clusters) two SPRY domains to bind the capsid simultaneously (*Goldstone et al., 2014*; *Javanbakht et al., 2007*; *Yap et al., 2007*). In principle, higher-order assembly of any geometry can amplify the recognition and restriction activities of the minimal dimer unit, but complete avidity and full restriction seem to occur when the TRIM5 dimers are arranged to match the lattice symmetry of the capsid. This principle appears exemplified by the L105A and W117A B-box mutants (*Table 2*). Like WT, the L105A mutant was fully restriction competent, bound to CA tubes efficiently in vitro, and formed observable hexagonal decorations on the tubes. In contrast, the W117A mutant was impaired in all three activities, even though it can clearly assemble spontaneously into higher-order but non-hexagonal arrays in vitro.

### TRIM lattice flexibility and curvature

Previous studies have analyzed how idealized (flat) capsid and TRIM lattices align in projection (*Ganser-Pornillos et al., 2011*; *Goldstone et al., 2014*; *Weinert et al., 2015*). This is reasonable because facets of the capsid surface are approximately flat, and because flat TRIM5α lattices assemble on two-dimensional crystals of the HIV-1 capsid protein. However, retroviral capsids present highly curved surfaces for TRIM5 binding, and the TRIM lattice must accommodate this curvature. Our structures demonstrate that the B-box/coiled-coil interface acts as a ball-and-socket joint, which imparts considerable flexibility in the way the coiled-coil arms emanate from the three-fold symmetric vertexes. Interestingly, our models define two major trajectories along which the coiled-coil can swing relative to the trimer plane (approximately parallel and approximately perpendicular). Motions parallel to the trimer plane change the angle between adjacent coiled-coils emanating from a vertex. This is consistent with the observation by *Li et al. (2016)* that the vertex angles in TRIM hexagonal lattices can deviate significantly from the ideal value of 120°, even in flat lattices. In our models, the coiled-coil can make an arc of about 12° along this trajectory. Angles between coiled-coil arms would therefore range from 120° ± 24°, which accounts for the full range of observed values (100–144°) (*Li et al., 2016*). Perpendicular motions, on the other hand, modulate the concavity of the triskelion and can generate lattice curvature. *Figure 10E and F* illustrate how the structural flexibility described above can allow the assembling TRIM lattice to scan for the most favorable SPRY domain positions and optimize local binding interactions, while simultaneously following the curvature of the bound capsid.

Flexibility in the triskelion arms also suggests a straightforward mechanism for generating multi-layered TRIM assemblies, because a coiled-coil can extend a B-box 2 domain above or below its current lattice plane to nucleate a new lattice. TRIM5α lattices in vitro are frequently multilayered, whether spontaneously assembled or nucleated by capsid templates (*Ganser-Pornillos et al., 2011*; *Li et al., 2016*; and this study). We speculate that the so-called cytoplasmic bodies that form upon overexpression of TRIM5α in cells (*Stremlau et al., 2004*; *Campbell et al., 2007*) might also assemble in this manner.

### Flexibility in B-box/B-box interactions

Our finding that miniTRIMs can form quasi-equivalent dimers and trimers uncovers yet another mode of flexibility in TRIM5α self-assembly. B-box dimers are clearly disfavored under our 'ideal' in vitro conditions wherein full-length TRIM5α assembles into hexagonal arrays. In principle, however, B-box dimers provide a means for extending the TRIM array in regions where the local assembly environment disfavors trimers. Indeed, apparent dimer linkages can be occasionally discerned in TRIM-coated capsids in vitro (*Li et al., 2016*). Interestingly, B-box dimerization is associated with local bending of the first two turns of the coiled-coil helix, as well as altered packing of the B-box

against the coiled-coil domain. This suggests a potential mechanism of allosteric communication that can link B-box oligomerization and coiled-coil dynamics. Indeed, our observation that the L105A mutant could not assemble into flat arrays while remaining competent in forming curved arrays is another indication of a structural and functional link between B-box/B-box interactions, the coiled-coil dimer scaffold, and lattice curvature.

B-box mediated interactions also promote dimerization of the upstream RING domain and E3 ligase activity (*Ganser-Pornillos et al., 2011*; *Pertel et al., 2011*; *Yudina et al., 2015*). The catalytically active RING configuration is more structurally compatible with a B-box dimer, and it is therefore possible that the B-box switches its oligomeric configuration to facilitate enzymatic function. In principle, it is possible for a RING-containing subunit to act simultaneously as ligase and ubiquitination substrate within the same E2-Ub/E3 complex, as biochemically demonstrated for the RING domain of RNF4 (*Plechanovová et al., 2011*). Nevertheless, our modeling studies also support a mechanism by which the B-box trimer could contribute to TRIM5α self-ubiquitination. Specifically, a B-box trimer could bring together three RINGs, with the first two acting as the E3 ligase and the third acting as the substrate. More important, our analysis further suggests that folding of the RING/B-box linker into a 4-helix bundle restricts the RING dimer to the outer surface of the TRIM hexagonal lattice, i.e., on the opposite side of the capsid binding surface. We suggest that this spatial compartmentalization facilitates formation of exposed polyubiquitin chains that can recruit downstream cytosolic factors to promote restriction and/or interferon signaling.

## Materials and methods

### MiniTRIM construction, expression, and purification

Synthetic DNA (Genewiz, Inc., South Plainfield, NJ) encoding the RBcc miniTRIM sequence in *Figure 1—figure supplement 1* and upstream His-tag and yeast Smt3p (SUMO) leader sequence was subcloned into pET30a (Novagen/EMD Millipore, Germany). To create the Bcc miniTRIM plasmid, the RING domain open reading frame (residues 1–88) was excised using a PCR-based linearization and religation protocol. Point mutations were introduced using the Quikchange method (Agilent, Santa Clara, CA). All plasmid constructs were confirmed by sequencing with T7 and/or T7 terminator primers.

Transformed *E. coli* BL21(DE3) cells were grown in LB broth supplemented with appropriate antibiotics and 50 µM zinc acetate. Cultures were shaken at 250 rpm and 37°C until the $OD_{600}$ reached 0.8–1.0. The shaker was then cooled to 18°C during induction with 1 mM isopropyl β-D-1-thiogalactopyranoside (IPTG). Cells were harvested by centrifugation 4 hr after induction then stored at −80°C.

Frozen *E. coli* weighing 25–30 g were resuspended in 120 mL of 2× lysis buffer (100 mM Tris, pH 8, 100 mM LiCl, 10% (v/v) glycerol, 1% (v/v) Triton X-100, 20 mM β-mercaptoethanol (βME), 2 mM phenylmethanesulfonylfluoride (PMSF)) then lysed using a microfluidizer (Microfluidics, Westwood, MA). The lysate was diluted to 1× with 120 mL cold water. Cell debris was pelleted by centrifugation at 45,000 *g* and discarded. The supernatant was then incubated with nickel agarose beads (Qiagen, Germany). The beads were washed with 10 column volumes (CV) of Wash 1 buffer (50 mM Tris, pH 8, 50 mM LiCl, 10 mM βME, 5% (v/v) glycerol), 2 CV of Wash 2 buffer (Wash 1 + 1 M LiCl), and again with 5 CV of Wash 1. Proteins were eluted by addition of 5 mL fractions of elution buffer (Wash 1 + 250 mM imidazole). The His-tag and SUMO leader sequences were cleaved off with SUMO-specific Ulp1 protease (3 µg/mL), during overnight dialysis in Wash 1 buffer. The His-SUMO protein was removed by a 15 min incubation with nickel agarose. The sample was then diluted 1.5× with water, and then applied to a HyperD anion exchange column (Pall Lifesciences, Port Washington, NY). Bound fractions were eluted with a linear gradient from 100% Wash 1 buffer to 70% Wash 1/ 30% Wash 2. Fractions were combined and concentrated to 0.5 mL then purified to homogeneity by gel filtration on a Superdex 75 column (GE Healthcare, Little Chalfont, UK) in 10 mM Tris, pH 8, 100 mM LiCl, 1 mM TCEP. Major peak fractions were pooled and concentrated to 3–15 mg/mL, flash-frozen in liquid nitrogen, then stored at −80°C. Typical yields were around 0.3 mg per L of culture for RBcc and around 1 mg per L for Bcc.

## Crystal structure determination

Protein stock solutions for crystallization trials generally consisted of about 3 mg/mL Bcc miniTRIM in 10 mM Tris, pH 8, 100 mM LiCl, 1 mM TCEP. Crystallization was performed in hanging drop format. Initial hits were identified with commercial sparse matrix screens. Optimized conditions are summarized in *Supplementary file 1A*. Diffraction data were collected at beamlines 22BM or 22ID at the Advanced Photon Source, and processed using HKL2000 (*Otwinowski and Minor, 1997*). We initially determined the structure of a dimeric Bcc miniTRIM (P6222 form) to 2 Å resolution by molecular replacement with a computational model derived from the rhesus TRIM5α B-box/coiled-coil structure (PDB 4TN3) (*Goldstone et al., 2014*) and residues 49–79 of *Thermus thermophilus* seryl-tRNA synthetase (PDB 1SRY) (*Fujinaga et al., 1993*). This model was partially refined and then used as a molecular replacement search model for all the other structures (*Table 1*). Structure determination and refinement were performed using the Phaser/AutoMR and phenix.refine modules of the PHENIX suite of programs (version 1.9–1692) (*Adams et al., 2010*). Secondary structure hydrogen bonding restraints and zinc coordination (bond and angle) restraints were used during refinement. Torsion angle (local) non-crystallographic symmetry (NCS) restraints were also used when appropriate. Manual model building was performed with the program Coot (*Emsley et al., 2010*). Structure validation tools, as implemented in both PHENIX and Coot were used throughout the structure refinement process.

## SEC-MALS

Mass measurements on Bcc miniTRIM were performed on a Dionex UltiMate3000 HPLC system with a UV detection module (ThermoFisher, Waltham, MA), connected to a miniDAWN TREOS static light scattering detector (Wyatt Technology, Santa Barbara, CA) and Optilab T-rEX differential refractometer (Wyatt Technology). A sample volume of 40 µL at 0.9 mM concentration was applied to a Superdex 200 HR 10/300 GL column (GE Healthcare) and developed in 30 mM Tris, pH 8.0, 100 mM NaCl at a flow rate of 0.4 mL/min. Data were recorded and processed using ASTRA software (Wyatt Technology).

## Analytical ultracentrifugation

Equilibrium sedimentation AUC experiments on Bcc miniTRIM were performed at 4°C using either XL-A or XL-1 analytical ultracentrifuges with absorbance optics (Beckman Coulter, Brea, CA). Sample cells with a six-channel centerpiece were filled with 110 µL of the protein samples at concentrations of 60, 30, and 15 µM, while 120 µL of sample buffer was loaded into the reference sectors. Absorbance scans at 280 nm were taken after equilibrium was reached (~12 h) at 14,000, 21,000, and 26,000 rpm. Protein partial specific volume and solvent density were calculated using SEDNTERP (version 20120828 BETA) available online at sedenterp.unh.edu. These values were used during curve fitting and data analysis using Heteroanalysis Software (version 1.1.58) (*Cole, 2004*).

## Restriction assays

Restriction assays were performed using HeLa cells grown in DMEM (Gibco/Thermofisher, Waltham, MA) supplemented with 10% fetal calf serum (Gibco) at 37°C in 5% $CO_2$. The cells were first transduced with a VSV-G pseudotyped lentiviral vector encoding rhesus TRIM5α with a C-terminal Flag-One-Strep (FOS) tag followed by an IRES sequence and DsRed (CSII-IDR2-TRIM5α-FOS). Three days after transduction, cells expressing TRIM5α-FOS were re-seeded in 24-well plates and transduced with increasing MOI of VSV-G pseudotyped HIV-GFP. A sample of these cells were pelleted and resuspended in SDS-PAGE sample buffer for western blot analysis of TRIM5α expression using anti-FLAG M2 antibody (Sigma, St. Louis, MO). 72 hr after HIV-GFP transduction, cells were trypsinized and analyzed for GFP expression (to determine the extent of HIV-GFP infection) and DsRed (as a marker for TRIM5α positive cells) by flow cytometry.

Lentiviral vectors for expressing TRIM5α in the above experiments were produced in the following manner. HEK293T cells were plated in 6-well plates and transfected with 1 µg pCMV-delR8.2, 0.4 µg pCMV-VSV-G, and 1.0 µg of the CSII-IDR2-TRIM5α-FOS plasmid. At 18 hr post-transfection, cells were placed in fresh media. At 48 hr post-transfection, media (containing the lentiviral particles) was removed and placed directly on HeLa cells for transduction and expression of TRIM5α-FOS and DsRed.

VSV-G pseudotyped HIV-GFP virions were produced in the same way as the TRIM5α expressing virions, but media harvested at 48 hr post-transfection was filtered through a 0.45 μ filter, layered on a 20% sucrose cushion in HS buffer (10 mM HEPES, pH 7.4, 140 mM NaCl) and centrifuged for 2 hr at 28,000 rpm in a Beckman SW32 Ti rotor. After centrifugation, the pellet containing viral particles was resuspended in HS buffer, aliquoted, and frozen for storage at −80°C. The number of infectious units was determined by titrating an aliquot on HeLa cells, and determining the fraction of HIV-GFP infected cells by flow cytometry three days later.

## TRIM5-21R mutagenesis, expression, and purification

The open reading frame from a previously described plasmid clone of Strep-FLAG-tagged TRIM5-21R (*Ganser-Pornillos et al., 2011*) was transferred into pFastBac1 (Invitrogen, Carlsbad, CA). Mutations were made in this vector using the Quikchange method (Agilent). Baculoviruses were made using a modification of the Invitrogen Bac-to-Bac system (*Hanson et al., 2007*). Proteins were expressed by infecting SF9 cells in shake culture format for 48 h, and purified as described previously (*Ganser-Pornillos et al., 2011*).

## TRIM5-21R assembly assays

Spontaneous TRIM5-21R assembly was achieved by incubation of purified protein samples at 4°C, as described (*Ganser-Pornillos et al., 2011*). The templated assembly assay for flat TRIM5-21R arrays was performed as described (*Ganser-Pornillos et al., 2011*). Co-assembly of TRIM5-21R and HIV-1 CA was performed as described (*Li et al., 2016*).

## Electron microscopy

Grid preparation and projection imaging of samples made by the spontaneous or template driven assembly methods were performed as described (*Ganser-Pornillos et al., 2011*). Electron tomography was performed as follows. Samples were mixed with BSA Gold Tracer (10 nm, Electron Microscopy Sciences, Hatfield, PA) and applied to glow-discharged continuous carbon grids for 1 min, washed with 0.1 M KCl, and stained with 2% (w/v) uranyl formate for 30 s. Tilt series were collected manually from –60° to +60° with a Tecnai F20 transmission electron microscope (Philips/FEI, Hillsboro, OR) operating at 120 kV. Step sizes were 5° at low tilts (0° to 30°), 3.5° at medium tilts (30° to 40°), and 1° at high tilts (30° to 60°). Images were recorded on a Gatan Ultrascan 4k × 4k CCD camera at a magnification of 29,000× (3.7 Å/pixel) and defocus of –2.5 μm. Images were aligned and binned by 4 using the IMOD software package (*Kremer et al., 1996*), and reconstructions were calculated with 16 iterations of the simultaneous iterative reconstruction technique using the TOMO3D software package (*Agulleiro and Fernandez, 2011*). Individual *z*-sections were visualized using the slicer option in IMOD. To boost the contrast and make features more visible, up to 5 sequential slices were combined.

## Capsid binding assays

The centrifugation assay was based on previously published methods (*Ganser-Pornillos et al., 2011*; *Stremlau et al., 2006*), with modification (*Fribourgh et al., 2014*). Disulfide-stabilized HIV-1 CA tubes (1 mg/mL, 20 μL) were incubated with TRIM5-21R (1 mg/mL, 5 μL) in ice for 1 hr. A 5-μL aliquot was removed, then mixed with equal volume of 2x SDS-PAGE sample buffer (load sample). The remaining sample was centrifuged at 16,000 *g* for 30 min at 4°C. The supernatant (20 μL) was removed and mixed with equal volume of 2x SDS-PAGE sample buffer. The pellet was resuspended in 40 μL of 1x SDS-PAGE sample buffer. Samples were boiled and then analyzed by SDS-PAGE (well volumes: 10 μL of load, 20 μL each of supernatant and pellet fractions) with Coomassie staining. Stained gels were scanned and quantified using ImageJ (*Schneider et al., 2012*). Binding activity was expressed as the fraction of TRIM5-21R in the pellet. By this method, we found that about 50% of freshly purified TRIM5-21R was consistently found in the pellet (*Figure 5*). Upon storage, the co-pelleted fraction dropped to about 30% (not shown). Mutants were therefore analyzed fresh after purification, two or three a time in parallel with a WT control.

## Molecular modeling

Model ensembles were calculated with the program RANCH as described (*Bernadó et al., 2007*), using the miniTRIM trimer structure from this study. Additional input files for the RING monomer and dimer ensembles, respectively, were the NMR structure of the RING monomer (PDB 2ECV) (*Lienlaf et al., 2011*) and the crystal structure of the RING dimer subunit (PDB 4TKP) (*Yudina et al., 2015*). Residues linking the RING and B-box domains were modeled as impenetrable spheres, with a $C\alpha$ angle distribution consistent with disordered proteins (*Bernadó et al., 2007*). Thus, possible interactions with the linker were disregarded. Since RANCH cannot model symmetry mismatched PDB files, the RING dimer orientations were modeled in the following manner. An ensemble of 30,000 models was calculated in the same manner as the RING monomer. The models were then computationally filtered to identify ones that contained a pair of RING domains wherein the distance of separation between two of the zinc lobes and their corresponding C-termini were consistent with the crystal structure of the RING dimer (*Yudina et al., 2015*). The RING dimer structure was then re-aligned manually, and then rotated to avoid steric clashes. We found that the RING 4-helix bundle was an important restraint that led to identification of only a handful of plausible RING dimer/B-box trimer configurations. The model for the self-ubiquitination complex was created by superimposing one of the RING dimer/miniTRIM trimer models, the crystal structure of the TRIM5$\alpha$ RING dimer in complex with Ubc13 (*Yudina et al., 2015*), and the structure of the RNF4 RING dimer in complex with ubiquitin-conjugated Ubc5a (*Plechanovová et al., 2012*). To model the third, substrate RING, the ensemble of 10,000 RING monomer/miniTRIM trimer models was then screened computationally to identify an orientation that placed the appropriate RING lysine (Lys45 or Lys51) within 8 Å of the E2-ubiquitin thioester bond without steric clashes between any of the structural elements.

## Accession numbers

Coordinates and structure factors are available from http://www.rcsb.org: P212121 trimer, 5IEA; C2 dimer, 5EIU; P1 dimer, 5F7T.

# Acknowledgements

We thank members of the Pornillos, Ganser-Pornillos, and Sundquist labs for experimental support and/or critical reading of the manuscript. Electron microscopy data were collected at the Molecular Electron Microscopy Core facility at University of Virginia. X-ray diffraction data were collected at beamlines 22-BM and 22-ID at the Advanced Photon Source, Argonne National Laboratory. Crystal screening was also performed with the assistance of B. Sankaran through the Collaborative Crystallography Program, Lawrence Berkeley National Laboratory at the Advanced Light Source. This study was supported by a seed grant from the Annette Lightner Foundation (OP) and NIH grants R01-GM112508 (OP) and P50-GM082545 (WIS and BKG-P). JMW was supported by a postdoctoral NIH fellowship (F32-GM115007). MDR participated in this study while on leave from Technical University of Lodz, Poland.

## Additional information

### Competing interests

WIS: Reviewing editor, *eLife*. The other authors declare that no competing interests exist.

### Funding

| Funder | Grant reference number | Author |
|---|---|---|
| National Institutes of Health | R01 GM112508 | Owen Pornillos |
| National Institutes of Health | P50 GM082545 | Barbie K Ganser-Pornillos Wesley I Sundquist |
| National Institutes of Health | F32 GM115007 | Jonathan M Wagner |
| Annette Lightner Foundation | | Owen Pornillos |

The funders had no role in study design, data collection and interpretation, or the decision to submit the work for publication.

## Author contributions

JMW, BKG-P, WIS, OP, Conception and design, Acquisition of data, Analysis and interpretation of data, Drafting or revising the article; MDR, KS, Conception and design, Acquisition of data, Analysis and interpretation of data; SLA, DC, Acquisition of data, Analysis and interpretation of data, Drafting or revising the article; GD, YW, Acquisition of data, Analysis and interpretation of data; GAF, Analysis and interpretation of data, Drafting or revising the article, Contributed unpublished essential data or reagents

## Author ORCIDs

Wesley I Sundquist, http://orcid.org/0000-0001-9988-6021
Owen Pornillos, http://orcid.org/0000-0001-9056-5002

# Additional files

## Supplementary files

• Supplementary file 1. (A) Crystallization conditions for Bcc miniTRIM. (B) Average root mean square deviations over equivalent Cα atoms from superpositions of the miniTRIM monomer structures with 4TN3 (*Goldstone et al., 2014*).

## Major datasets

The following datasets were generated:

| Author(s) | Year | Dataset title | Dataset URL | Database, license, and accessibility information |
|---|---|---|---|---|
| Jonathan M Wagner, Ginna Doss, Owen Pornillos | 2016 | TRIM5 B-box2 and coiled-coil chimera | http://www.rcsb.org/pdb/explore/explore.do?structureId=5IEA | Publicly available at the Protein Data Bank (accession no. 5IEA) |
| Jonathan M Wagner, Ginna Doss, Owen Pornillos | 2016 | Mini TRIM5 B-box 2 dimer C2 crystal form | http://www.rcsb.org/pdb/explore/explore.do?structureId=5EIU | Publicly available at the Protein Data Bank (accession no. 5EIU) |
| Jonathan M Wagner, Ginna Doss, Owen Pornillos | 2016 | TRIM5 B-box2 and coiled-coil chimera | http://www.rcsb.org/pdb/explore/explore.do?structureId=5F7T | Publicly available at the Protein Data Bank (accession no. 5F7T) |

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
