## [Decision Letter]

Congratulations: we are very pleased to inform you that your article, "Mechanism of higher-order assembly of the HIV restriction factor TRIM5α", has been accepted for publication in *eLife*. The Reviewing Editor for your submission was Stephen Goff and the Senior Editor was Wenhui Li.

Both our reviewers were supportive about the paper. One had several suggestions for improvement, and I think all of these are good points – I think making most of the changes as suggested will improve the clarity.

Reviewer #1:

Most TRIM5 expression constructs generate insoluble products that have made structural assessment difficult. Some success has been had with subdomains, but understanding how the domains fit together has been elusive. To address the core TRIM5 assembly, the authors generated B-box domain constructs (RBcc miniTRIM and Bcc miniTRIM) with a truncated coiled-coil domain and hairpin linker that behaved well in solution. R121E mutants disrupted the ability of the miniTRIMs to oligomerize, as shown by size exclusion chromatography. Crystals were obtained with Bcc miniTRIM and multiple structures were solved that overlapped with a previous structure of the B-box 2/coiled-coil dimer. The miniTRIM formed dimers and trimers, the latter with 3 B-box 2 domains at the vertex and a contribution from the N-terminus of the coiled-coil helix. Most were dimers in solution, though the trimer form better matched the hexamer models observed with full-length protein. Mutations that disrupted critical interactions (E102K, K103E, E102K/K103E, E120R, and R121E) were generated in the context of full-length protein and shown to disrupt restriction activity in tissue culture. Expression levels correlated inversely with the ability to assemble and to restrict. All TRIM5 mutants were able to pellet with CA assemblies albeit with reduced efficiency. Mutations that reduced restriction activity (L106A, W117E, L118D, and L132D) prevented or decreased spontaneous assembly of TRIM5-21R into a hexameric lattice, or assembly with CA. The authors then generated models to explain how TRIM5 forms a lattice that accommodates binding to the variable curvature of the retroviral CA lattice, formation of B-box trimers, RING dimers bound to Ub-E2s, as well as proximity of a RING domain K51 to the thioester of Ub-E2. The creative approaches here have paid off. They are beautifully executed and greatly advance our understanding of the structure/function of TRIM5. The manuscript is well written, clear.

Reviewer #2:

This manuscript from Wagner and colleagues describes the results of experiments aimed at understanding the higher order assembly of TRIM5α on the surface of retroviral capsid cores. TRIM5α is a cytoplasmic restriction factor that recognizes the mature capsid core after viral entry and prevents completion of the first half of the viral replication cycle. TRIM5α contains an N-terminal RING domain, a B-box domain, a coiled coil and a C-terminal PRYSPRY domain. The protein dimerizes via the coiled-coil domain forming an extended linear structure, with the two PRYSPRY (recognition) domains coming together at the center of the dimer. Recent published work has shown that recombinant TRIM proteins representing TRIM5α appear to assemble as a hexameric lattice on the surface of the viral capsid lattice, and that each side of a TRIM hexamer is composed of a dimer. The current paper focuses on understanding multimeric interactions between B-box domains, and suggests that these trimerize to form the three-fold vertices of the TRIM5 lattice. The results also suggest that the TRIM5α lattice has flexibility at the vertices allowing concavity necessary to coat a curved surface (such as the icosahedral capsid shell) as well as flexibility in the angle between the TRIM5α "arms" extending from the B-boxes.

All experiments focus on "miniTRIMs", recombinant proteins that contain a B-box and a short segment of the coiled coil held in place by a heterologous coiled-coil hairpin from a bacterial protein. While an alternative structure that also contained a RING domain was made, it failed to crystallize and is not characterized in the study.

The authors are appropriately cautious about the use of the heavily modified proteins, and there are no major concerns. The implied model including formation of the vertices, a mechanism for allowing concavity, and placement of the RING domains above the TRIM5 plane (and away from the capsid), is logical and should inspire additional work by this and others. The model is also consistent with/supported by the co-submission by Li et al.

Specific points:

The title should refer to "retrovirus restriction factor" rather than "HIV restriction factor"; the latter implies some sort of specificity, and more importantly, detracts from the very interesting fact that TRIM5α recognizes a conserved higher-order structure rather than a virus-specific epitope or motif. The title maybe also promises more than is in the paper – the focus is exclusively on the B-box, and maybe this would be better specified in the title?

Results – "our miniTRIM constructs are excellent surrogates for the head regions of the full RBCC motif" is misleading, in particular since not much information is presented for the RBcc miniTRIM.

In the subsection “Structure-based TRIM5α B-box 2 mutants are impaired in restriction and capsid binding”, third paragraph – in the absence of quantitation, the section suggesting an inverse correlation between expression levels and assembly seems like a bit of a stretch. Are there data available to bolster this conclusion, or alternatively, could the authors make it clear that this touches on speculation?

Figure 1 – schematic would be easier to read if the CC and linker derived domains were somehow labeled; including some indication of where the SPRY domains "would" be (light gray ovals, or dashed lines, or asterisks?) would also be useful for the Discussion. In (E), why aren't the R121E derivatives also shown in the gel?

Figure 2 – the blue spheres aren't labeled or mentioned in the legend.

Figure 3 – asterisks aren't explained.

Figure 3—figure supplement 3 – very minor – the symbols are so closely overlapping they are hard to make out – might help to either put each label close to the corresponding "curve", or to add a color for each symbol.

Figure 3—figure supplement 5 – what is the arrow?

Figure 4, indicating that the panels are top layer interactions and bottom layer interactions right in the figure and legend would make it easy to follow.

Figure 5 – what is the unexplained band in panel B?

Figure 9 – also minor, mostly to aid the reader, is it possible to indicate likely dimensions of the underlying capsid lattice? Or at least in panel B, indicate a capsid layer below the figure?

---

## [Author Response]

Reviewer #2:

Specific points:

The title should refer to "retrovirus restriction factor" rather than "HIV restriction factor"; the latter implies some sort of specificity, and more importantly, detracts from the very interesting fact that TRIM5α recognizes a conserved higher-order structure rather than a virus-specific epitope or motif. The title maybe also promises more than is in the paper – the focus is exclusively on the B-box, and maybe this would be better specified in the title?

We have changed the title to “Mechanism of B-box 2 domain mediated higher-order assembly of the retroviral restriction factor TRIM5α”.

Results – "our miniTRIM constructs are excellent surrogates for the head regions of the full RBCC motif" is misleading, in particular since not much information is presented for the RBcc miniTRIM.

The phrase (subsection “Bcc miniTRIM recapitulates the B-box/coiled-coil head of TRIM5α”) now states: “our miniTRIM constructs are excellent structural surrogates for the B-box/coiled-coil core of the full RBCC motif.”

In the subsection “Structure-based TRIM5α B-box 2 mutants are impaired in restriction and capsid binding”, third paragraph – in the absence of quantitation, the section suggesting an inverse correlation between expression levels and assembly seems like a bit of a stretch. Are there data available to bolster this conclusion, or alternatively, could the authors make it clear that this touches on speculation?

We now emphasize that this is a speculation that explains the correlation. The last two sentences of this paragraph now state: “We therefore speculate that the ability to assemble reduces steady state protein levels because TRIM5α proteins that assemble are turned over more rapidly in cells. An extension of this argument is that the ability of TRIM5α to assemble correlates with its ability to restrict HIV-1."

Figure 1 – schematic would be easier to read if the CC and linker derived domains were somehow labeled; including some indication of where the SPRY domains "would" be (light gray ovals, or dashed lines, or asterisks?) would also be useful for the Discussion. In (E), why aren't the R121E derivatives also shown in the gel?

Panel B has now been modified and the domains labeled. Panel E now shows all 4 miniTRIM constructs.

Figure 2 – the blue spheres aren't labeled or mentioned in the legend.

The blue zinc atoms are now mentioned in the legend.

Figure 3 – asterisks aren't explained.

This was briefly mentioned in the main text (subsection “Structures of the Bcc miniTRIM trimer and dimer”, first paragraph). We have now included a brief explanation in the figure caption.

Figure 3—figure supplement 3 – very minor – the symbols are so closely overlapping they are hard to make out – might help to either put each label close to the corresponding "curve", or to add a color for each symbol.

The figure is now in color.

Figure 3—figure supplement 5 – what is the arrow?

The caption now explains that the arrow indicates the direction of the “clam shell-like” motion.

Figure 4, indicating that the panels are top layer interactions and bottom layer interactions right in the figure and legend would make it easy to follow.

We agree and have made this change.

Figure 5 – what is the unexplained band in panel B?

The band is an apparent proteolytic product of TRIM5-21R, and this is now noted in the caption.

Figure 9 – also minor, mostly to aid the reader, is it possible to indicate likely dimensions of the underlying capsid lattice? Or at least in panel B, indicate a capsid layer below the figure?

We agree that this change makes the figure more informative and have added the capsid layer to panel B.